# Bond-selective intensity diffraction tomography

Jian Zhao [1,7,9], Alex Matlock [1,8,9], Hongbo Zhu [2] ✉, Ziqi Song[3], Jiabei Zhu[1], Biao Wang[2], Fukai Chen[4], Yuewei Zhan [5], Zhicong Chen[1], Yihong Xu [6], Xingchen Lin[2], Lei Tian [1,5] ✉ & Ji-Xin Cheng [1,5,6] ✉

Recovering molecular information remains a grand challenge in the widely used holographic and computational imaging technologies. To address this challenge, we developed a computational mid-infrared photothermal microscope, termed Bond-selective Intensity Diffraction Tomography (BS-IDT). Based on a low-cost brightfield microscope with an add-on pulsed light source, BS-IDT recovers both infrared spectra and bond-selective 3D refractive index maps from intensity-only measurements. High-fidelity infrared fingerprint spectra extraction is validated. Volumetric chemical imaging of biological cells is demonstrated at a speed of ~20 s per volume, with a lateral and axial resolution of ~350 nm and ~1.1 μm, respectively. BS-IDT's application potential is investigated by chemically quantifying lipids stored in cancer cells and volumetric chemical imaging on *Caenorhabditis elegans* with a large field of view (~100 μm x 100 μm).

Optical microscopy plays a pivotal role in modern biological research and clinical practice[1]. Its capabilities of visualizing and quantifying subcellular structures provide deep insights into cell physiology. Among various solutions, quantitative label-free microscopy has gained popularity from being able to investigate biological objects in their native state, thereby circumventing fluorescence microscopy's weaknesses including phototoxicity, photobleaching, and cellular functions perturbations[2]. Several label-free microscopy methods based on elastic scattering, such as holographic imaging[3–8] and computational imaging methods[9–14], have been implemented to recover subcellular morphology. These methods provide high-speed quantifications of the objects' optical phase delay or refractive index (RI) distributions with nanometer resolution and nanoscale sensitivity and have gained significant progress in applications such as living neuron activity evaluations[15], cell mass quantifications[16], mitotic chromosomes characterizations[17], and volumetric tissue histopathology[18]. These

solutions, however, are fundamentally limited by their lack of molecular specificity, preventing the differentiation of biochemical compositions or subcellular structures with similar morphologies.

To realize chemical-specific label-free microscopy, vibrational spectroscopic imaging techniques have been developed to chemically image cellular morphology based on signals from intrinsic chemical bond vibrations[19,20]. Among numerous technical realizations, coherent Raman scattering microscopy has been developed for high-speed vibrational imaging and applied to address various biomedical problems[21–23]. Despite its success, Raman scattering is a weak scattering process with an extremely small scattering cross-section (~$10^{-30}$–$10^{-28}$ cm²). In most cases, coherent Raman scattering imaging requires tightly-focused laser beams with large excitation power, resulting in a high potential for photodamage[24]. In comparison, IR absorption offers a cross-section (~$10^{-18}$ cm²) that is ten orders of magnitude larger than Raman scattering[25]. Furthermore, IR imaging

[1]Department of Electrical and Computer Engineering, Boston University, Boston, MA 02215, USA. [2]State Key Laboratory of Luminescence and Applications, Changchun Institute of Optics, Fine Mechanics and Physics, Chinese Academy of Sciences, Changchun 130033, China. [3]Aerospace Information Research Institute, Chinese Academy of Sciences, Beijing 100190, China. [4]Department of Biology, Boston University, Boston, MA 02215, USA. [5]Department of Biomedical Engineering, Boston University, Boston, MA 02215, USA. [6]Department of Physics, Boston University, Boston, MA 02215, USA. [7]Present address: The Picower Institute for Learning and Memory, Massachusetts Institute of Technology, Cambridge, MA 02142, USA. [8]Present address: Department of Mechanical Engineering, Massachusetts Institute of Technology, Cambridge, MA 02142, USA. [9]These authors contributed equally: Jian Zhao, Alex Matlock. ✉e-mail: zhbciomp@163.com; leitian@bu.edu; jxcheng@bu.edu

can be implemented without a tight beam focus, featuring higher chemical sensitivity and reduced photodamage risk. The emerging mid-infrared photothermal (MIP) microscopy[19] inherits IR absorption spectroscopy's advantages but circumvents conventional IR micro-spectroscopy's low-resolution and slow speed limit. MIP microscopy provides diffraction-limited resolution at the visible band using a visible probe beam and is compatible with both point-scanning[26–29] and wide-field[30–35] configurations. Yet, existing MIP microscopy suffers from slow volumetric imaging speed and low depth resolution.

More recently, the MIP effect has been harnessed to bring molecular specificity to holographic microscopy and realize MIP-based high-performance quantitative volumetric chemical imaging. In this direction, several interferometry-based holographic chemical imaging methods have been proposed[31,33,34]. Zhang et al. demonstrated the first MIP holographic microscope enabling 2D quantitative chemical imaging on unlabeled living cells[31]. However, this method only partially recovers the complex biological process, and valuable depth-resolved information is still missing. Recently, Ideguchi's team reported MIP 3D holographic microscopy using optical diffraction tomography (ODT) for depth-resolved chemical cellular imaging[33]. This approach unravels the phase information from interferometrically modulated scattered light fields. However, the modality requires a complicated optical illumination beamline, a two-arm interferometer, and specialized optics for implementation. These features tend to increase phase noise, optical misalignment, and mechanical instabilities, which limit the detection sensitivity and the system compatibility with commercial microscopes. In practice, this method requires a significant amount of averaging to achieve an adequate signal-to-noise ratio (SNR) and limits the acquisition to ~12.5 min per volume. The demonstrated depth resolution for this approach is limited to approximately 3 μm with a field of view (FOV) comparable to a single cell. These limitations hinder the full exploration of volumetric chemical imaging and negate the high-speed advantages of wide-field illumination configurations.

In this work, we present a non-interferometric computational MIP microscopy scheme for 3D bond-selective label-free imaging. Our scheme enables both high-resolution, high-speed volumetric quantitative chemical imaging and high-fidelity mid-infrared fingerprint spectroscopy within a standalone imaging modality. Our method synergistically integrates the time-gated pump-probe MIP microscopy with the pulsed laser-based Intensity Diffraction Tomography (IDT), termed Bond-Selective Intensity Diffraction Tomography (BS-IDT). The time-gated pump-probe detection captures the transient 3D RI variations at a microsecond timescale. The 3D RI is quantitatively measured by the pulsed IDT using a scan-free, non-interferometric setup. Notably, our system is built as an add-on to a commercial brightfield microscope to significantly reduce system complexity. The scan-free and common-path design minimizes mechanical instabilities and phase noise. These unique features allow us to realize high-speed (~0.05 Hz, up to ~6 Hz) and high-resolution (~350 nm laterally, ~1.1 μm axially) 3D hyperspectral imaging with a large FOV (~100 μm × 100 μm). Compared to the state-of-the-art ODT-based MIP microscopy[33], our BS-IDT improves the quantitative chemical volumetric imaging speed by ~40 times, depth resolution by ~3 times, and FOV by ~3 times. These unique capabilities of BS-IDT are shown in four demonstrative experiments. We first demonstrate high-fidelity recovery of mid-IR fingerprint spectroscopic information enabled by BS-IDT. Next, we highlight BS-IDT's high-speed chemical imaging capabilities on single-cell samples and recover mid-IR fingerprint spectra focusing on protein and lipid bands. We further validate BS-IDT's application potential by quantitatively extracting 3D chemical information from cell organelles. In addition, we show that our system can also achieve 2D bond-selective differential phase contrast (BS-DPC) computational microscopy and highlight the benefits of the 3D imaging capability of BS-IDT through a quantitative comparison of the two methods on the same cell samples. Finally, we showcase large FOV chemical imaging on a

multicellular *Caenorhabditis elegans* (*C. elegans*) object as well as mid-IR fingerprint spectra extractions.

## Results

### BS-IDT principle, instrumentation, and image reconstruction
BS-IDT integrates the IDT modality with a pump-probe MIP wide-field detection scheme to provide chemical information with high temporal and spatial resolution. The mid-IR pump laser triggers MIP effects in the sample, while the IDT component provides an easily implementable imaging system probing the MIP-induced chemical-specific 3D RI variation. To account for the temporal constraints of MIP microscopy's pump-probe detection, we developed a pulsed IDT system with a customized nanosecond (ns) pulsed laser ring array to capture these RI variations. The principle, instrumentation, and image reconstruction of BS-IDT are detailed below.

As illustrated in Fig. 1a, BS-IDT utilizes MIP effects from the mid-IR fingerprint region (~5 μm to ~20 μm) and a pump-probe detection technique to generate a chemical-specific RI map. Each absorption peak area in the mid-IR fingerprint region corresponds to a unique molecular vibrational bond that, if harnessed, can differentiate distinct biochemical compounds in the sample[36]. When a mid-IR laser pump beam illuminates a sample, the radiation absorbed by the molecular vibrational bond causes a transient and localized temperature increase. This MIP-induced sample expansion modifies the sample's local RI distribution, leading to the sample's scattering cross-section variations[37]. The generated heat can dissipate within a few microseconds to tens of microseconds[37,38]. The pump mid-IR pulse is oscillated between "On" and "Off" at high speed, creating periodic mid-IR light absorption in the sample. This oscillation creates "Hot" and "Cold" states, respectively, where the chemical-specific RI variations are present or absent in the sample. The sufficiently fast and sensitive pulsed IDT imaging system with multiple off-resonant probe beams can capture this information within a microsecond timescale to recover the chemical-specific RI variations of the object quantitatively by subtraction between "Hot" and "Cold" states[19,39].

Capturing this transient RI fluctuation requires a unique pump-probe pulsed IDT imaging system. Figure 1b shows a schematic of the complete BS-IDT system (see Methods for details). The main block of this system consists of a brightfield transmission microscope created using low-cost, off-the-shelf optical elements and a reflective lens for focusing the mid-IR pump beam. The main unique element in BS-IDT is a lab-built laser ring array containing 16 low-cost continuous wave (CW) diode lasers with a 450 nm central wavelength that obliquely illuminate the sample for the IDT probe illumination. A pulsed laser array, instead of the LED array in conventional IDT[10], is required to synchronize with the high speed of the pulsed mid-IR beam for detecting the MIP-induced RI variations. Thus, each CW diode laser is electrically modulated at a tunable repetition rate (0 to 10 kHz) and pulse duration (~0.6 μs to ~1 μs). During data acquisitions, each diode laser is operated at the same repetition rate and pulse duration as the mid-IR laser for both the "Hot" and "Cold" states. The oblique illumination angle of each laser is set to match the microscope's objective numerical aperture (NA), which maximizes the spatial frequency coverage allowed by the system. This spatial frequency enhancement follows synthetic aperture principles[40] and expands the accessible bandwidth to achieve the diffraction-limited resolution of incoherent imaging systems. To reduce spatial coherence and suppress speckle noise, we further installed diffusers at the output of each optical-fiber-coupled laser diode. For providing the mid-IR sample illumination, we install an off-axis gold parabolic mirror above the sample stage to integrate the MIP pump-probe detection into our BS-IDT setup. The pulsed mid-IR laser beam illuminates the sample under an on-axis configuration. The parabolic mirror focuses the mid-IR beam spot to a size of ≈63 μm Full width at half maximum (FWHM) to enhance the mid-IR laser intensity at the area of interest. This mid-IR beam size

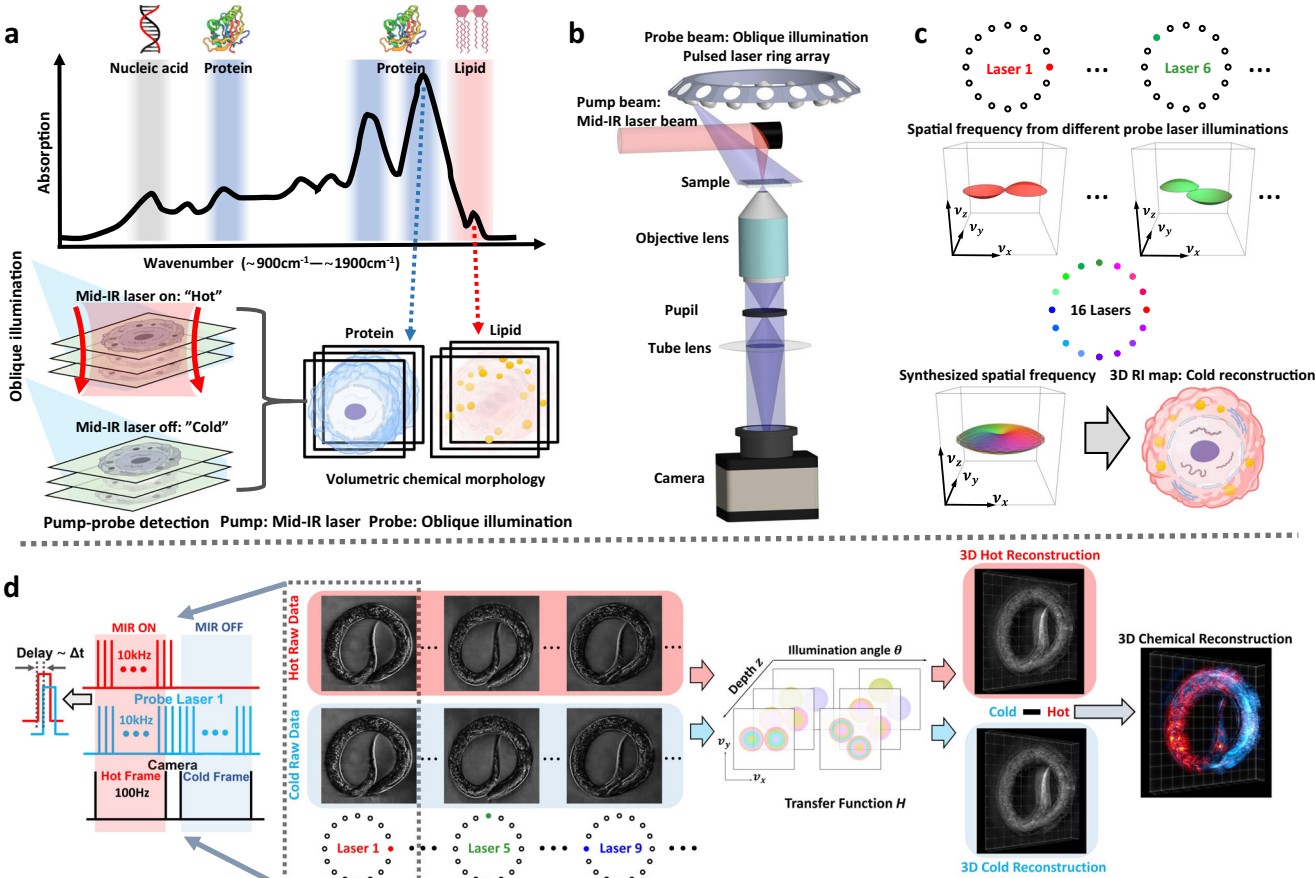

**Fig. 1 | BS-IDT principle, instrumentation, and image reconstruction. a** MIP-based chemical imaging principle. Created with BioRender.com. The top inset in "**a**" shows a mid-IR spectrum in the fingerprint region of ~900 cm⁻¹–1900 cm⁻¹. Each crest in the spectrum corresponds to a unique chemical compound. The sample under mid-IR laser illumination: "Hot" state; sample without mid-IR laser illumination: "Cold" state. Pump-probe detection: pulsed mid-IR laser illumination (pump beam) on the sample causes transient local RI change. RI variations for hot and cold states are probed by another visible oblique beam. 3D chemical morphology maps with different biomolecular distributions: (1) subtract hot reconstruction from the cold; (2) tune wavenumber to obtain a different chemical compound of interest. Biomolecule icons in "**a**" are created with ref. 65. Cancer Cell icon in "**a**" is adapted from ref. 65. **b** System design. BS-IDT is based on a wide-field transmission microscope (Methods). Probe beam: oblique laser illumination (~450 nm) from a laser array that consists of 16 diode lasers. Pump beam: off-axis gold parabolic mirror focuses mid-IR (MIR) laser beam from a quantum cascade laser (QCL) onto the

sample. **c** 3D RI map reconstruction. Created with BioRender.com. For each oblique illumination, the collected 2D intensity data can be mapped into the 3D frequency domain. Using data from all 16 oblique illuminations, the 3D cold RI map can be recovered by an inverse Fourier transform of the synthesized Ewald's sphere. Cancer Cell icon in "**c**" is adapted from ref. 65. **d** 3D chemical imaging workflow. The leftmost inset of "**d**" shows a time synchronization scheme. The pulse duration for both probe and MIR laser is ~1 μs. The probe pulse is ~0.5 μs delayed relative to the MIR pulse. For the MIR laser, an additional 50 Hz on/off duty-cycle modulation is imposed to generate "Hot" and "Cold" states. For each probe laser illumination, paired "Hot" and "Cold" 2D raw imaging data are collected as indicated by dashed-line square. "Hot" or "Cold" 3D RI maps are reconstructed using all the 16 "Hot" or "Cold" raw images based on the IDT forward model, illustrated as the angle-dependent depth-resolved phase transfer function. By subtraction operation, the 3D chemical image illustrating RI variations is extracted.

decides the chemical imaging FOV for a single wide-field measurement and is sufficiently large for encompassing single cells. For larger objects, such as *C. elegans*, we extend this FOV by scanning the mid-IR beam and stitching the chemical imaging information computationally. This BS-IDT design simplifies the system realization to the extent that a regular low-cost brightfield microscope can be upgraded to BS-IDT merely by replacing the illumination sources.

Based on the above BS-IDT's system design, we further illustrate the principle of the "Cold" 3D RI map reconstruction in Fig. 1c (see Methods for details). To reconstruct the 3D RI biological sample map, BS-IDT implements a physical model relating the objects' properties to the scattering information recorded by the intensity images[10,41–43]. Specifically, BS-IDT utilizes the first-Born approximation that models the scattering generated by the sample as a linear problem considering only the single scattering events between the incident field and the object. This approximation implies that the scattered field from each point throughout the object space is independent and allows the object to be considered as an axially discretized set of decoupled 2D

slices. This discretization enables slice-wise 3D recovery of the object's RI using an easily implementable, efficient, and closed-form deconvolution inverse method. As shown in Fig. 1c, the CMOS camera captures a 2D intensity image encoding the object's 3D volume for each oblique illumination. The cross-interference extracted from intensity images can be mapped into the 3D frequency domain. By synthesizing all the spectra data obtained from 16 different illuminations, the 3D object can be recovered by transforming the synthesized Ewald's sphere back to the spatial domain[43]. The recovered 3D RI map lays the foundation for MIP-based bond-selective volumetric imaging.

Lastly, a workflow for 3D chemical imaging is demonstrated in Fig. 1d. Time synchronization is crucial to capture the transient RI fluctuations. As shown in the leftmost inset of Fig. 1d, BS-IDT synchronizes the probe laser, the pump mid-IR laser, and the camera at an acquisition speed of 100 Hz. Both diode laser and mid-IR laser are modulated at a 10 kHz repetition rate and ~1 μs pulse duration. Each probe laser pulse is precisely synchronized with the corresponding mid-IR laser pulse with a short time delay. This time delay (~0.5 μs) is

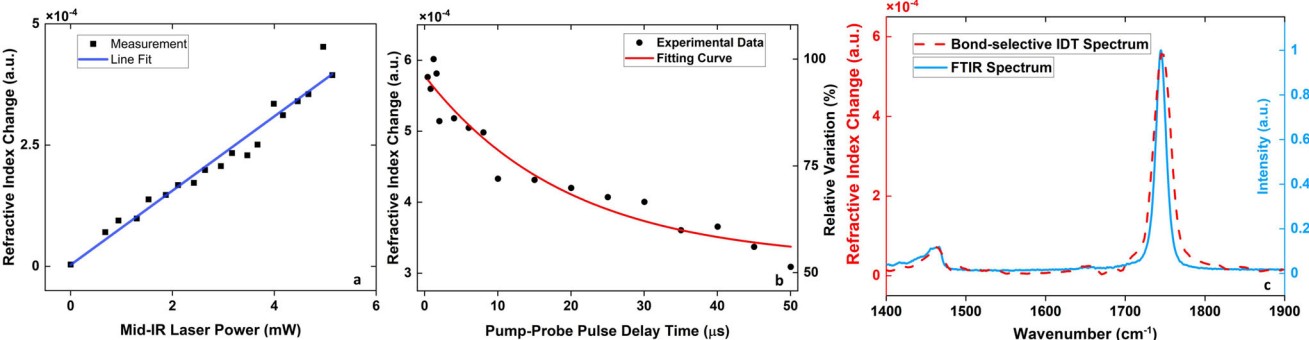

**Fig. 2 | BS-IDT system performance. a** Dependence of MIP-induced RI changes on mid-IR laser power and the linear fitting curve with a slope of $7.7 \times 10^{-5}$. **b** MIP-induced RI changes and relative RI change with respect to the pump-probe time delay variations. Relative RI variation is obtained by dividing the RI change over the maxima. The red fitting curve shows an exponential temporal decay constant of ~20 µs. **c** Spectral fidelity of BS-IDT validated by the standard FTIR spectrum of the same specimen. Sample: soybean oil film.

added to the probe laser pulse to capture the maximum RI variation from the sample. Since the CMOS camera is operated at 100 Hz, 50 Hz duty-cycle modulation is further imposed on the mid-IR laser pulse train to switch the recorded frame between "Hot" and "Cold" states. For wide-field MIP microscopy, the application of a pulsed probe laser here increases the contrast and sign-to-noise ratio, considering the slow frame rate of our camera[30,31]. For each probe laser illumination at a fixed pump mid-IR wavenumber, BS-IDT collects a pair of "Hot" and "Cold" raw images. For a complete volumetric imaging acquisition process, BS-IDT generates 16 "Hot" or "Cold" raw images. Following the acquisition, BS-IDT uses IDT's inverse scattering model[10] to reconstruct the "Hot" and "Cold" 3D RI structures using 32 raw images. A simple subtraction of the two reconstructed volumes reveals small (~$10^{-4}$ to ~$10^{-3}$) RI fluctuations due to the MIP-induced changes in the sample for a particular wavenumber. This process not only provides the 3D structure of the sample but also returns a volumetric molecular composition map throughout the object. Following a similar workflow, BS-IDT can provide site-specific mid-IR spectra from the hyperspectral 3D chemical maps by scanning the mid-IR wavenumber, uncovering various unique biochemical compound distributions. More importantly, the spectroscopic information enables the extraction of the fingerprint absorption spectrum from arbitrary volumetric areas of interest with unknown chemical compositions. Chemometric analysis further decodes the chemical information utilizing the extracted fingerprint spectrum, which is not feasible for fluorescence microscopy[36].

## BS-IDT system characterization

We characterized the system performance of BS-IDT and demonstrated the results in Fig. 2. Using soybean oil film as a testbed, we performed three independent tests quantifying the RI variation with respect to mid-IR laser power, delay time between the pump and probe laser pulses, and the mid-IR wavenumber. Here, the delay time is defined as the temporal shift between the probe pulse's center and the pump pulse's center. The sample was made by sandwiching an oil film uniformly distributed between two pieces of 0.2-mm-thick Raman-grade Calcium Fluoride (CaF$_2$) glass with a diameter of 10 mm. For the first two tests (Fig. 2a, b), we fixed the wavenumber of the mid-IR laser to 1745 cm$^{-1}$, corresponding to the C=O stretch vibration in the oil sample. We first characterized the dependence of RI variation on the mid-IR pump laser power, as shown in Fig. 2a. A clear linear relation exists between the mid-IR absorption and recorded RI changes (R square coefficient = 0.96). This result indicates no system-induced nonlinear errors in the measured RI variations based on the mid-IR pump power. Next, we quantified the RI variations with the pump-probe delay time (Fig. 2b). The experimental data are fitted by an exponential decay function with a temporal decay constant of ~20 µs. The thermal temporal decay constant varies with thermal diffusivity

and sample spatial size[37,38]. With a large area and size, the oil film sample shows a slower decay time than other smaller biological objects. We performed experimental measurement and numerical simulations for particles with sizes comparable to organelles in cells (Supplementary information: "Heat dissipation measurement and simulation"). These particles demonstrate much faster temporal decay processes compared to the oil film. Therefore, the temporal pulse spacing of our mid-IR laser is ~100 µs, allowing sufficient cooling. Finally, we compared BS-IDT's absorption spectroscopy with standard Fourier transform infrared (FTIR) Spectroscopy (Fig. 2c) to confirm that our system properly extracts the oil's spectrum. The FTIR spectra intensity is shown in blue, while the red dashed-line highlights the BS-IDT recovered RI variation. The close agreement in recovered spectral information verifies that BS-IDT adequately recovers the MIP-induced RI change. Together, these tests indicate that BS-IDT faithfully recovers the chemical-specific RI variations in a sample of interest.

## 3D chemical imaging, mid-IR fingerprint spectroscopy, and metabolic profiling of cancer cells

In Fig. 3, we demonstrate the 3D chemical imaging and mid-IR fingerprint spectroscopy capabilities of BS-IDT on fixed human bladder cancer cell samples. Human bladder cancer cells (T24 cells) were fixed, washed, and immersed in D$_2$O phosphate-buffered saline (PBS) between two pieces of 0.2-mm-thick Raman-grade CaF$_2$ glasses. D$_2$O was chosen for the immersion medium as it minimizes the water IR absorption and demonstrates a relatively flat spectral response[44]. To illustrate 3D chemical imaging, this sample was first imaged under different mid-IR laser illumination conditions (Fig. 3a–k) selected for protein, lipid, off-resonance imaging as well as imaging with the mid-IR laser beam blocked. Mid-IR fingerprint hyperspectral imaging was then performed on T24 cells to illustrate the spectroscopic capabilities of BS-IDT (Fig. 3l). Finally, BS-IDT's potential for cell metabolic profiling is shown by identifying cancer cells with differing invasiveness based on their intracellular lipid content (Fig. 3m).

Figure 3a–k illustrates the 3D chemical imaging capabilities of BS-IDT. Figure 3a–e shows individual slice-wise reconstructions of two mitotic T24 cells with five different conditions: (a) cold RI map of the entire cell structure, (b) Amide I band protein absorption map at 1657 cm$^{-1}$, (c) lipid C=O absorption map at 1745 cm$^{-1}$, (d) RI variation map at 1900 cm$^{-1}$ off-resonance where minimal RI variations should appear, and (e) RI variation map when the mid-IR pump beam is blocked. For each image, we averaged 60 camera frames for each diode laser illumination to generate chemical imaging results with a speed of ≈0.05 Hz acquisition rates. Figure 3a–c shows all the cellular features obtained in Fig. 3a at varying axial positions. Only specific structures contain protein (Fig. 3b) or lipids (Fig. 3c). The images from two control groups (Fig. 3d, e) confirm that the chemical imaging

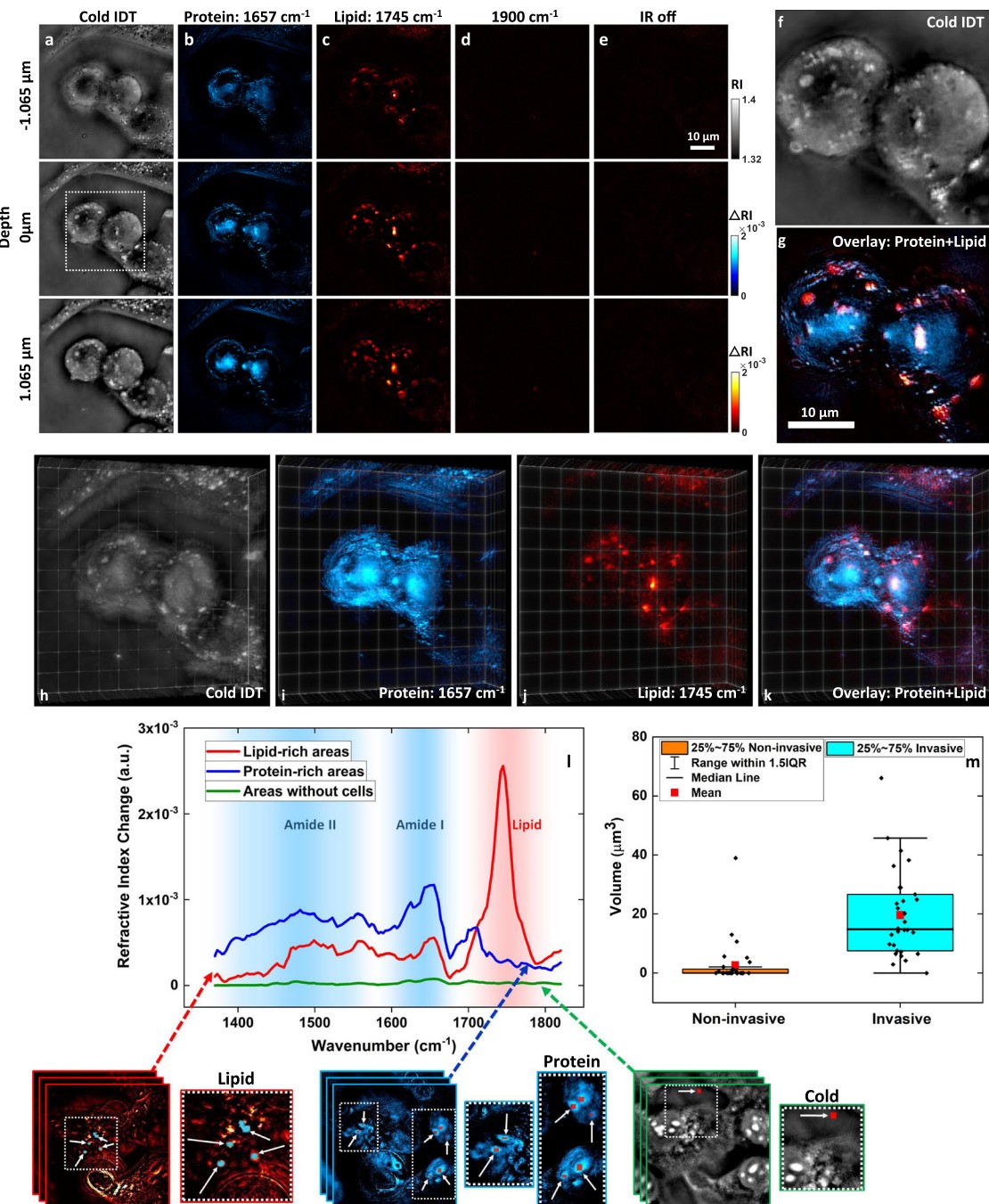

**Fig. 3 | BS-IDT imaging on bladder cancer cells. a–k** 3D chemical imaging of fixed bladder cancer T24 cells under mitosis. The medium is $D_2O$ PBS. The scale bar in "**e**" is applicable to "**a**" to "**e**". **a** Depth-resolved cold imaging results. **b–e** Depth-resolved chemical imaging results. For "**a**" to "**e**", images in each row are from the same depth. For "**b**" to "**e**", images in each column are under the same mid-IR laser illumination conditions. **b** Protein imaging results with 1657 cm$^{-1}$ mid-IR wavenumber. **c** Lipid imaging results with 1745 cm$^{-1}$ mid-IR wavenumber. **d** Control tests: off-resonance imaging results with 1900 cm$^{-1}$ mid-IR wavenumber. Most biochemical compounds have significantly weak or no absorptions at this wavenumber. **e** Control tests: chemical imaging results without mid-IR illumination.

**f** Cold imaging results of the dash-line square area in "**a**". **g** Overlay chemical imaging results of the dash-line square area in the dash-line square area of "**a**". The scale bar in "**g**" is for both "**f**" and "**g**". **h–k** 3D reconstructions of the cells shown in "**a**", "**b**", and "**c**". **l** Mid-IR Fingerprint spectra extracted from protein and lipid-rich areas as well as areas without cells. The chemical images below "**m**" show the areas of interest highlighted with different colors and indicated by white arrows.

**m** Statistical results of lipid content volume measured from T24 cells and SW780 cells. T24 is an invasive bladder cancer cell line. SW780 is a non-invasive bladder cancer cell line. 30 independent cells were used for each cell type. The volumes of lipids were independently measured and extracted 30 times per cell type.

contrast originates from MIP effects. The proteins appear to be relatively uniformly distributed throughout the cell's cytosol, while the lipids are mostly concentrated within lipid droplet organelles for energy storage. As many different cellular functions are underway during mitosis, we would expect a strong protein response throughout the cytosol. We characterized the system resolution using the lipid

features extracted from the absorption map at 1745 cm$^{-1}$ (Supplementary information: "BS-IDT resolution characterization"). The measured lateral and axial FWHM linewidths are ~349 nm and ~1.081 μm, respectively. When these chemical-specific images are overlaid (Fig. 3g, k), we observed that these protein and lipid structures are predominantly separate with only some overlap for specific lipid

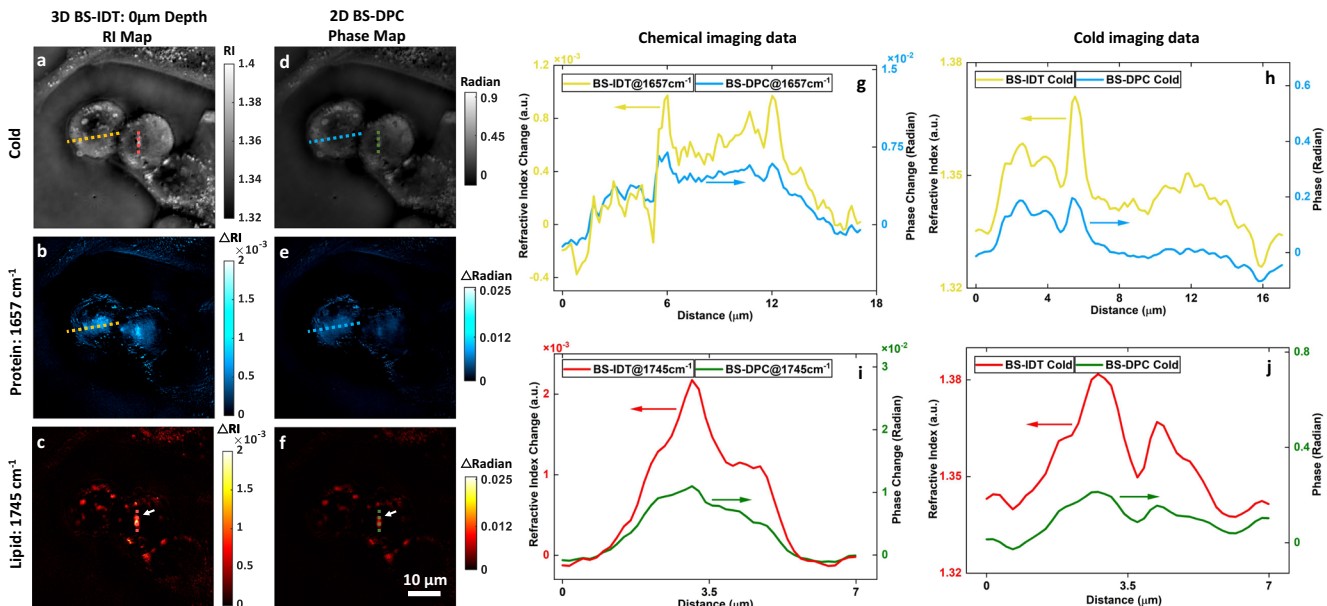

**Fig. 4 | Comparison of BS-IDT and BS-DPC microscopy. a–c** BS-IDT cold and chemical 3D RI reconstruction results sectioned at 0 μm depth. Sample: fixed mitotic bladder cancer cells in $D_2O$ PBS. **d–f** BS-DPC cold and chemical 2D phase reconstruction results of the same sample. For images in "**a–c**" and "**d–f**", each row corresponds to the same mid-IR radiation. **g, i** are line profiles extracted from the chemical imaging results of BS-IDT and BS-DPC. **g** Yellow-color profile line is extracted from the yellow dash-line area of "**b**". Blue-color profile line is extracted from the blue-dash-line area of "**e**". **i** Red-color profile line is extracted from the red dash-line area of "**c**". Green-color profile line is extracted from the green dash-line area of "**f**". **h, j** are line profiles extracted from the "cold" imaging results of BS-IDT and BS-DPC. **h** Yellow-color profile line is extracted from the yellow dash-line area of "**a**". Blue-color profile line is extracted from the blue-dash-line area of "**d**". **j** Red-color profile line is extracted from the red dash-line area of "**a**". Green-color profile line is extracted from the green dash-line area of "**d**".

droplets. When rendered in 3D (Fig. 3h–k, Supplementary Movie 1), we clearly see the 3D distribution of these chemical-specific structures. Adding this information to 3D RI reconstructions provides additional clarity regarding the cellular structures and drastically improves the wealth of information available with label-free imaging.

Next, we evaluate the mid-IR spectroscopy capabilities of BS-IDT using separate, non-mitotic T24 cells. We performed hyperspectral 3D chemical imaging on fixed bladder cancer cells to generate a stack of 3D chemical images. Each 3D chemical image corresponds to a certain mid-IR wavenumber point in the spectrum. We can extract the mid-IR fingerprint spectra from protein and lipid-rich areas averaged over a few selected voxels (Fig. 3l). As a control, we also extracted the mid-IR spectrum from the medium area without cell samples. The spectrum extracted from protein-rich areas shows the signature Amide I absorption peak ~1650 cm$^{-1}$. Due to the hydrogen-deuterium exchange[45], two crests (~1550 cm$^{-1}$ and ~1450 cm$^{-1}$) corresponding to the Amide II band can also be observed. For lipid-rich areas, a strong C=O signature peak ~1750 cm$^{-1}$ can be observed. As a comparison, the spectrum extracted from the liquid medium does not show strong absorption signatures. These spectral results further confirm that BS-IDT can properly extract chemical-specific information and identify specific spectroscopic structures from biological cell samples.

We further apply BS-IDT to identify the invasiveness of two types of bladder cancer cells by quantifying lipid droplet volumes. Prior work has shown that dysregulated lipid metabolism is a common characteristic of human cancer[46,47]. Lipid alteration is accepted as one of the biomarkers for potential cancer, and lipid droplet accumulation is related to metastatic cancer[46,48,49]. Statistical analysis of lipid droplet volume allows for quantitative evaluations of the lipid accumulations in cancer cells. However, prior methods rely on a complex indirect data extraction procedure from fluorescence microscopy imaging data[50]. With BS-IDT, we can quantify the lipid droplet volume directly from our 3D reconstruction images without requiring fluorescent labeling. Here, we used invasive T24 cells and non-invasive SW780 cells to illustrate BS-IDT's performance in lipid quantification. For each test

group, we measured 30 different cells at the 1745 cm$^{-1}$ wavenumber lipid absorption peak. Then, we calculated the total lipid droplet volume per cell and evaluated the data in a boxplot (Fig. 3m). We calculated the lipid droplet volume by converting the total voxels into true spatial dimensions (Supplementary information: "Lipid quantification"). Based on the lipid content volume distribution, there is a significantly higher level of lipid accumulation in invasive cancer cells than the non-invasive cell. This experiment provides additional proof that dysregulated cell metabolism accumulates more lipid content in cancer cells. It further demonstrates the application potential of BS-IDT as a label-free chemical quantitative method in biomedicine.

## Comparison between 3D BS-IDT and 2D bond-selective differential phase contrast imaging

To further illustrate the benefit of BS-IDT's 3D chemical imaging, we compared 2D and 3D reconstructions from the same dataset using the Differential Phase Contrast (DPC) model and the IDT model, respectively. DPC microscopy is a 2D, non-interferometric computational imaging technique that merely requires four oblique illuminations[51]. Despite lacking depth-resolved 3D imaging capabilities, DPC microscopy's minimal image requirement enables high-speed imaging to achieve the same incoherent diffraction-limited resolution as IDT. Bond-Selective DPC (BS-DPC) microscopy can be performed using the same hardware as IDT. For BS-DPC method details, please refer to the "Supplementary information" document. Here, both the BS-IDT imaging results and the BS-DPC imaging results obtained using the same raw bladder cancer cell dataset are demonstrated in Fig. 4. We show the cold (Fig. 4a, d), the protein band (Fig. 4b, e), and the lipid-band (Fig. 4c, f) imaging results, as well as the extracted line profiles (Fig. 4g–j). The BS-IDT recovers the RI map, while BS-DPC recovers the phase map. BS-DPC imaging allows high-quality bond-selective chemical imaging results but demonstrates lower contrast and degraded resolution compared to BS-IDT's imaging results. Despite that both methods use the same raw dataset, different algorithms affect the reconstructed SNRs. This degradation mainly originates from BS-DPC

method's integration of the cell's features across various depths. In contrast, BS-IDT's 3D imaging capability enhances the fine features and improves imaging contrast. In addition, BS-IDT is able to obtain temperature variations since it can de-couple RI from the optical path. For BS-DPC, directly extracting temperature change is still challenging in that the DPC method inherently recovers the integrated phase through the object volume.

### 3D chemical imaging and mid-IR fingerprint spectroscopy of multicellular organism

Observing the 3D molecular composition of multicellular organisms, like the *C. elegans* worm, can serve as an important model system to decipher many fundamental biology questions, including lipid metabolism and its connection to aging and disease[52,53]. However, evaluating such specimens requires a complex process with conventional methods using exogenous contrast agents or dye stains[53]. These approaches can often be detrimental to the sample and make it difficult to properly locate molecules of interest within the volumetric object[53]. Complex labeling protocols can also easily damage the sample and hinder biological research. Thus, visualizing the volumetric distributions of chemical bonds within such samples is highly desirable with a label-free method.

In Fig. 5, we illustrate multicellular organism 3D chemical imaging with BS-IDT on a *C. elegans* worm. The *C. elegans* (daf-2 (e1370) mutant strain) was fixed, immersed in $D_2O$ PBS, and sandwiched by two pieces of 0.2-mm-thick $CaF_2$ glass. For imaging this specimen, the IDT probe illumination captured the entire worm in a single measurement (Fig. 5a, d, g), but the focused IR beam only illuminated a subset of the worm at each position. To resolve this FOV mismatch, multiple IDT acquisitions were obtained while scanning the IR beam through all segments of the worm. For the ≈100 μm × 100 μm BS-IDT FOV, we acquired ten measurement sets while scanning the IR beam through the worm and computationally stitched them together in post-processing. Each scan took ~19.2 s per measurement. We repeated the above imaging process for different mid-IR wavenumbers to recover protein (Amide I band, 1657 cm$^{-1}$) and lipid (C=O band, 1745 cm$^{-1}$) 3D morphologies throughout the sample (Fig. 5b, c). With BS-IDT's 3D reconstruction capabilities, we can observe all object slices in a 3D rendering highlighting the chemical distribution throughout the sample (Fig. 5g–j, Supplementary Movie 2). Finally, we further validated the mid-IR spectroscopy capabilities on this type of multicellular sample through a spectroscopic scan on a portion of the *C. elegans* worm (Fig. 5k).

Our *C. elegans* imaging results highlight the significant potential for this modality in evaluating complex multicellular specimens. From the slice-wise reconstructions and rendering in Fig. 5, we observe spatial dependency in the lipid and protein distribution within *C. elegans*. Amide I protein band resonance appears strongest in the worm's anterior half, while significant lipid storage exists along the digestive tract towards the worm's posterior in concentrated circular structures in 3D space. This distribution agrees with our expectations, as the worm stores lipids within fat granules towards its posterior when it is well-fed[54]. In the zoom-in tail region (Fig. 5d–f), we observe that the chemically-sensitive BS-IDT can separate the lipid granules and proteins that exhibit similar spherical geometries. This added detection provides significant benefits for biologists wanting label-free analysis of the *C. elegans* structures. Through the spectroscopic analysis in Fig. 5k, we further confirm these signatures are truly proteins and lipids, and there exist unique mixtures in varying structures within the sample. Evaluating the composition and ratios of lipids to proteins in these structures could provide new insights into *C. elegans* development with broader impacts on other biological research fields. These results show the exciting potential for BS-IDT in imaging larger complex biological specimens with molecular specificity.

## Discussion

BS-IDT realizes high-speed (~0.05 Hz, up to ~6 Hz) and high-resolution (~350 nm laterally, ~1.1 μm axially) 3D chemical-specific, quantitative computational imaging over a large FOV (~100 μm × 100 μm) and mid-IR fingerprint spectroscopy on cells and multicellular *C. elegans* with a simple system design. BS-IDT has improved the chemical volumetric imaging speed by ~40 times, the depth resolution, and the FOV by ~3 times, as compared to the state-of-the-art interferometric ODT-based MIP method[33].

BS-IDT's superior performance can be attributed to several innovations in its instrumentation and advances in computational imaging. First, BS-IDT provides high-resolution 3D chemical images by a unique pump-probe pulsed IDT design. IDT utilizes oblique illuminations encoding high spatial frequency information about the sample into the microscope's passband up to the incoherent resolution limit. By further using the 450 nm short-wavelength probe beam to capture the MIP-induced RI variations, BS-IDT achieves high-resolution chemical imaging while bypassing the low-resolution restrictions of conventional IR methods like FTIR micro-spectroscopy[55]. Benefiting from the wide-field imaging scheme with a fast-diverging illumination design, the probe beam intensity incident on the sample is ~2 × 10$^{-6}$ mW/μm$^2$, eight orders of magnitude lower than Raman[56] or coherent Raman microscope[57]. Furthermore, the use of the IDT modality allows for the chemical phase signal to be decoupled into height and RI from its 3D inverse scattering model, which cannot be achieved with the previous MIP 2D holographic microscopy methods[31,34]. The end result of BS-IDT provides richer information regarding RI variations and the structural distribution of the sample's chemical composition. Second, BS-IDT's high-speed chemical 3D imaging originates from its effective system design and efficient computational algorithms. The programmable, electrically scanned laser array provides fast illumination scanning without mechanical motion. This scan-free configuration, together with the non-interferometric imaging system design, minimizes the amount of image averaging required for noise suppression. IDT's algorithm further boosts the imaging speed due to its highly efficient linear inverse scattering model. Third, BS-IDT's simple modular design provides a universal and scalable chemical imaging platform. BS-IDT requires no specialized optics and can be adapted to a regular brightfield microscope as an add-on module. The decoupled reflective pump beamline can also be scalable to applications using pump lasers beyond the mid-IR spectrum region so as to add greater chemical detection capabilities. This beamline design natively enables a ~60 μm FOV, sufficient for single-cell studies and spectroscopic cell profiling, as shown in our studies on bladder cancer cells. In addition, the FOV can be easily expanded by performing independent IR beam scanning to achieve multicellular organism scale FOV chemical imaging, as demonstrated in the *C. elegans* imaging with a FOV reaching ~100 μm × 100 μm.

The performance of BS-IDT can be further enhanced with the following future advances. On the laser source side, the low mid-IR pulse energy of QCL is the bottleneck of the MIP chemical signal. Tens of nanosecond and high-energy solid-state lasers, such as pulsed mid-IR optical parametric oscillators, would provide two-fold benefits for BS-IDT imaging: (1) Significant enhancement of the chemical signal to noise ratio and (2) Larger FOV using a weakly focused mid-IR beam. On the computational imaging aspect of the system, IDT's physical model cannot be reliably applied to multiple-scattering samples without incurring greater error[18]. It is highly desired to introduce multiple-scattering IDT models[11,58] into the current framework to extend the scope of BS-IDT to image strongly scattering biological samples, such as thick tissues, which will open up many other exciting biomedical applications. In addition, the BS-IDT could potentially deploy the visible probe illumination under CW mode to simplify the timing scheme further. To this end, an ultrafast camera with a frame rate matching the repetition rate of the pump laser pulse is required.

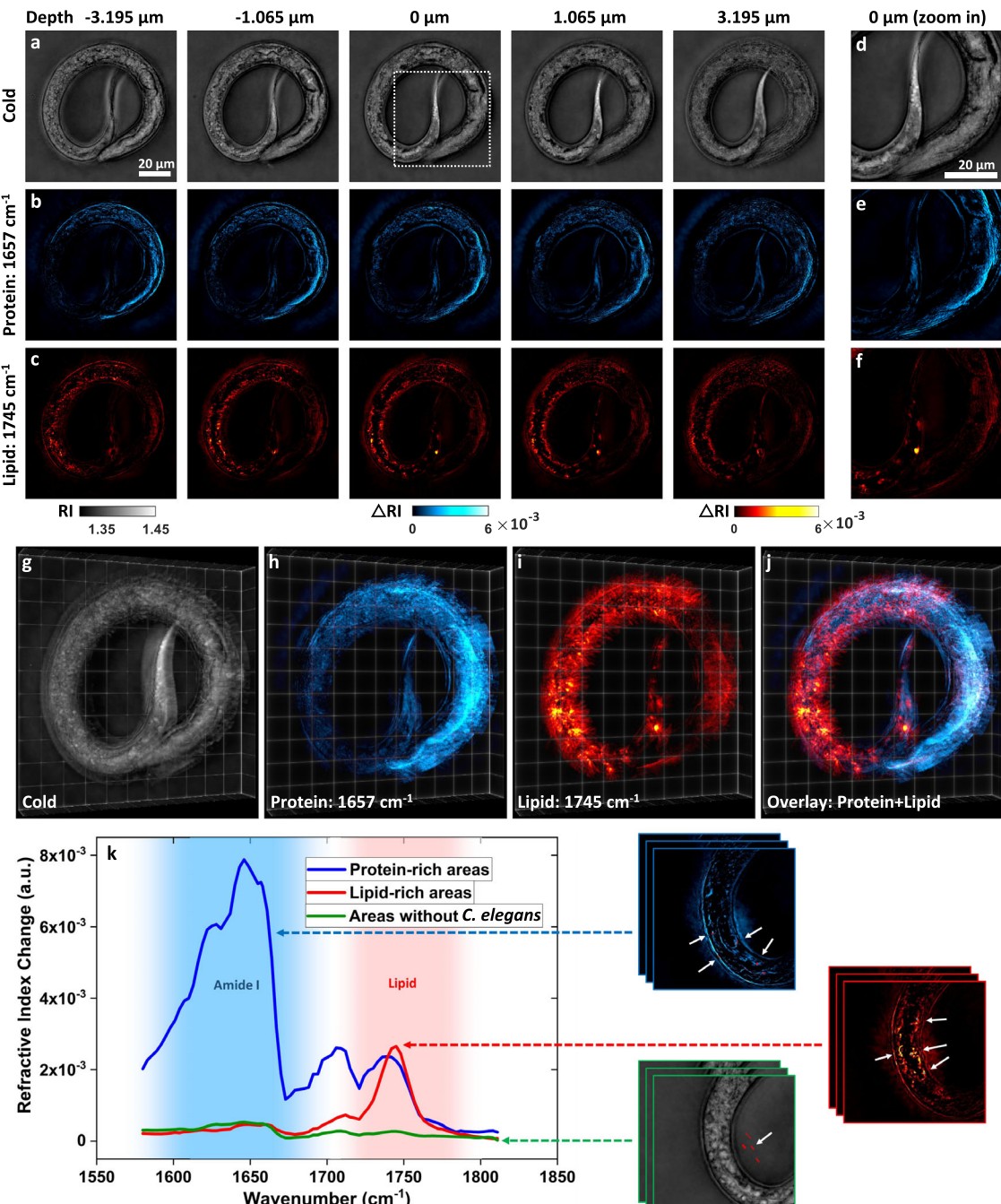

**Fig. 5 | BS-IDT imaging and fingerprint spectroscopy on *C. elegans*. a–j** Sample: fixed *C. elegans* (daf-2 (e1370) mutant strain) immersed in $D_2O$ PBS. From "**a**" to "**f**", images in each column are from the same depth. Images in each row are under the same mid-IR laser illumination conditions. The scale bar in "**a**" is applicable to "**a**–**c**". The scale bar in "**d**" is applicable to "**d**–**f**". **a, d** Depth-resolved cold imaging results. **b, e** Depth-resolved protein imaging results at 1657 cm⁻¹ wavenumber. **c, f** Depth-resolved lipid imaging results at 1745 cm⁻¹ wavenumber. **d–f** Cold, protein, and lipid imaging results of dash-line square area in "**a**". **g–i** 3D rendering of the *C. elegans* shown in "**a**", "**b**", and "**c**". **j** is the 3D overlay of "**h**" and "**i**". **k** Mid-IR Fingerprint spectra extracted from protein and lipid-rich areas as well as areas without worms. The chemical images on the right side of the plot show the areas of interest that are used to extract the spectra.

BS-IDT's quantitative 3D chemical-specific computational imaging capabilities provide significant application potential in biological imaging. First, BS-IDT could provide an alternative to FTIR microspectroscopy with 3D quantitative information and orders of magnitude better resolution. Second, BS-IDT can non-destructively quantify the volume and mass of different chemical components distribution or organelles inside a single cell without needing contrast agents. In this work, we have already demonstrated the BS-IDT's capabilities in evaluating lipid contents' volume. This capability is crucial in investigating some of the most important cell biology questions, such as the growth

regulation of mammalian cells[59]. Third, BS-IDT visualizes 3D morphology and structure of cells or organelles with molecular specificity in a label-free manner that enables essential applications for understanding various diseases. For example, providing 3D visualizations of Tau protein aggregates in neural tissue could contribute to the understanding of the mechanism of neurodegenerative diseases, such as Alzheimer's disease[60]. Conventional fluorescence microscopy perturbs the cell functions and cannot provide quantitative 3D visualization of the intrinsic protein aggregates[61]. BS-IDT could, therefore, play an important role in these areas. Fourth, BS-IDT can be applied to IR metabolic

dynamic imaging of living organisms, covering single-cell-level and tissue-level metabolic activity profiling. Such metabolic imaging can be implemented with the assistance of the IR-active vibrational probes and by extending the spectral coverage to the cell-silent window (1800 cm$^{-1}$ – 2700 cm$^{-1}$). For example, IR tags, such as the azide group, $^{13}$C-edited carbonyl bond, and deuterium-labeled probes[25], can be exploited to investigate various metabolic activities simply by adding another QCL laser module to cover both the fingerprint region and the cell-silent region. BS-IDT imaging of IR tags with a sub-micrometer resolution, high speed, and rich spectral information would contribute to unraveling the mechanisms of many biological processes[25,62]. Fifth, BS-IDT can also be deployed in material characterizations. For many applications, chemical microparticle characterizations require non-contact, stain-free, and sub-micrometer resolution detection[63]. BS-IDT could find new applications in these areas. Finally, the modular design of BS-IDT ensures high compatibility with different pump-probe microscopy schemes. Thus, our design is not limited to the mid-infrared regime. As an independent optical module, photothermal effects beyond the mid-IR band can also be introduced.

In summary, BS-IDT demonstrates superior performance using simple and low-cost solutions, providing an attractive label-free solution to realize 3D chemical-specific, quantitative computational microscopy. BS-IDT's high-speed, sub-micrometer volumetric chemical imaging capabilities provide invaluable information for understanding basic science and various diseases. The large FOV chemical imaging capability demonstrates the BS-IDT's utilities from single cells to multicellular objects. Notably, these advances are based on a modular and cost-effective design developed from a regular bright-field microscope. We envision that the BS-IDT technology can be widely adopted and deployed in a broad range of applications.

## Methods

### Instrumentation

The brightfield microscope for BS-IDT consists of a microscope objective (Thorlabs, RMS40x,0.65 NA, 40x magnification), a tube lens (Thorlabs, TTL180A, f = 180 mm), a silver mirror (Thorlabs, PF20-03-P01), and a CMOS camera (Andor ZYLA-5.5-USB3-S). The microscope optics were assembled using a standard Thorlabs 60 mm cage system. For the ring laser illumination system, the array consists of 16 individual diode lasers (wavelength: ~450 nm, average power under CW mode: ~3 W, repetition rate: up to 10 kHz, pulse duration: ~0.6 μs to ~10 μs). For all the chemical imaging experiments, the probe laser output power under pulsed mode is ~30 mW based on the ~0.01 duty cycle. The probe beam was coupled and transmitted through multimode optical fibers (0.22 NA, 105 μm core diameter). The probe beam illumination area on the sample has a diameter of around 4 cm. We customized a ring fiber head holder that guarantees the illumination angle matches the microscope objective's NA. Each optical-fiber head is designed as an instant plug-in for the ring holder. The ring holder can either be made with metallic materials in a machine shop or 3D printed with plastic materials. This holder can be modified to incorporate additional diode lasers or to provide different illumination angles matching the NA of other microscope objectives. We also customized a set of circuit boards and a microcontroller to control the 16 diode lasers. Each diode laser is easy to plug-in/pull out from the circuit boards. The mid-IR pump laser is a Daylight solution MIRcat-2400 QCL laser. A gold parabolic mirror (Thorlabs, MPD01M9-M03, Reflected focal length: 33 mm) re-directs the mid-IR beam into the sample. For pump-probe detection, the pulse duration and repetition rate are set to ~1 μs and 10 kHz for both probe and pump beam. The energy fluence of the probe beam on the sample area is ~0.2 pJ/μm$^2$. The mid-IR energy fluence on the sample is ~50 pJ/μm$^2$, depending on the wavenumber. The Andor camera runs at 100 Hz frame rate during data acquisition. In order to synchronize the probe pulse, the pump pulse, and the camera frame rate, we use a

pulse generator (Quantum Composers 9214) to synchronize the timing and control the pump-probe pulse delay. In addition, we apply duty cycle control to the mid-IR laser trigger signal so that 10 kHz mid-IR laser pulse train is turned on and off at 50 Hz.

### Imaging model

BS-IDT utilizes the conventional intensity diffraction tomography model originally published by Ling et al.[42] and the annular IDT work by Li and Matlock et al.[10] for recovering the 3D RI distributions of the sample. We briefly review this model here and direct the readers elsewhere for additional detail[10,41,42]. For BS-IDT, we model the object as a 3D scattering potential within a given volume $\Omega$ as $V(\mathbf{r},z) = k^2(4\pi)^{-1}\Delta\varepsilon(\mathbf{r})$, where $\mathbf{r}$ denotes the 3D spatial coordinates $\langle x,y,z \rangle$, $k$ is the probe beam wavenumber, and $\Delta\varepsilon(\mathbf{r})$ is the permittivity contrast between the object and the imaging medium. Each oblique laser illumination on the sample acts as a plane wave $u_i(\mathbf{r}|\mathbf{v}_i)$ incident on the sample at a given angle defined by its lateral spatial frequency vector $\mathbf{v}_i$. Under the first-Born approximation, the model assumes the total field generated from this incident field scattering from the object can be evaluated as a summation

$$u_{tot}(\mathbf{r}|\mathbf{v_i}) = u_i(\mathbf{r}|\mathbf{v_i}) + \int_\Omega u_i(\mathbf{r'}|\mathbf{v_i})V(\mathbf{r'})G(\mathbf{r} - \mathbf{r'})d^3\mathbf{r'}, \quad (1)$$

of the incident and first-order scattered field defined by a 3D convolution with the Green's function $G(\mathbf{r})$. The IDT model assumes that the total scattered field from the object results from a stacked set of 2D axial slices through the object because the scattering events from each sample point are mutually independent. This assumption implies that the object's volumetric distribution can be recovered from a single 2D plane if the additional propagation is included in the inverse model for recovering each axial slice.

To recover the 3D object, BS-IDT relates the object's volumetric scattering potential to the system's measured intensity images using the cross-interference between the incident and scattered field. This cross-interference linearly encodes the object's scattering potential into intensity. Coupled with oblique illumination, the cross-interference term and its conjugate are spatially separated in the Fourier plane allowing for linear inverse scattering models under weakly scattering assumptions. With this separation and the further assumption that the object's permittivity contrast is complex ($\Delta\varepsilon(\mathbf{r},z) = \Delta\varepsilon_{re}(\mathbf{r},z) + j\Delta\varepsilon_{im}(\mathbf{r},z)$), a forward model relating the background-subtracted image intensity spectra to the volumetric object can be developed

$$\hat{I}(x,y|\mathbf{v_i}) = \sum_m H_{re}(\mathbf{v},m|\mathbf{v_i})\Delta\hat{\varepsilon}_{re}(\mathbf{v},m) + H_{im}(\mathbf{v},m|\mathbf{v_i})\Delta\hat{\varepsilon}_{im}(\mathbf{v},m), \quad (2)$$

where $\hat{\cdot}$ denotes the Fourier transform of a variable, $m$ denotes the axial slice index, and $H_{re}$ and $H_{im}$ are the transfer functions (TFs) containing the physical model. These TFs have the form

$$H_{re}(\mathbf{v},m|\mathbf{v_i}) = \frac{jk^2\Delta z}{2}A(\mathbf{v_i})P(\mathbf{v_i})\left[P(\mathbf{v}-\mathbf{v_i})\frac{e^{-j[\eta(\mathbf{v}-\mathbf{v_i})-\eta(\mathbf{v_i})]m\Delta z}}{\eta(\mathbf{v}-\mathbf{v_i})} - P(\mathbf{v}+\mathbf{v_i})\frac{e^{j[\eta(\mathbf{v}+\mathbf{v_i})-\eta(\mathbf{v_i})]m\Delta z}}{\eta(\mathbf{v}+\mathbf{v_i})}\right], \quad (3a)$$

$$H_{im}(\mathbf{v},m|\mathbf{v_i}) = -\frac{k^2\Delta z}{2}A(\mathbf{v_i})P(\mathbf{v_i})\left[P(\mathbf{v}-\mathbf{v_i})\frac{e^{-j[\eta(\mathbf{v}-\mathbf{v_i})-\eta(\mathbf{v_i})]m\Delta z}}{\eta(\mathbf{v}-\mathbf{v_i})} + P(\mathbf{v}+\mathbf{v_i})\frac{e^{j[\eta(\mathbf{v}+\mathbf{v_i})-\eta(\mathbf{v_i})]m\Delta z}}{\eta(\mathbf{v}+\mathbf{v_i})}\right], \quad (3b)$$

where $A(\mathbf{v_i})$ denotes an illumination source amplitude, $P(\mathbf{v})$ is the microscope's circular pupil function, $\Delta z$ is the discretized slice thickness, and $\eta(\mathbf{v}) = \sqrt{\lambda^{-2}-|\mathbf{v}|^2}$ is the axial spatial frequency with imaging wavelength $\lambda$ and translation dependent on the illumination angle. Given this linear forward model, the inversion of this model is straightforward using a slice-wise deconvolution with Tikhonov regularization.

## BS-IDT mid-IR spectroscopy

To extract the mid-IR fingerprint spectrum, we first reconstructed the 3D chemical images for each wavenumber. For spectral fidelity validation, we used soybean oil film and selected areas of interest arbitrarily. The values of selected voxels were averaged to obtain one point for the corresponding wavenumber. We repeated this procedure to obtain the raw spectrum. Then, we normalized the raw spectrum with the measured mid-IR laser intensity spectrum and further denoised the results using a Savitzky-Golay filter[64]. We used an FTIR spectrometer (Bruker Vertex 70 v FTIR spectrometer) to measure the ground-truth soybean oil film mid-IR spectrum. We followed a similar procedure for the mid-IR spectrum extractions of cells and worms.

## BS-IDT image stitching method

When evaluating the juvenile *C. elegans*, a total of ten BS-IDT measurement sets were acquired for each wavenumber to scan the IR beam throughout the worm's entirety. During the reconstruction, these images required stitching to form a continuous chemical response throughout the worm. Conventional stitching methods such as alpha blending are not viable for this process, as the Gaussian profile of the IR beam generates a corresponding Gaussian chemical signal response within each worm section. To ameliorate this issue, we performed a Gaussian blending process to stitch the chemical signatures together.

In the blending process, a separate intensity image set was first acquired from each IR position using a red laser (~633 nm) illumination that propagates along the same beam path as the pump laser. This illumination acts as a guide star providing the central position of the mid-IR illumination. Using this guide, a centroid position was estimated from the guide star to approximate the centroid for the IR laser illumination. Using the IR laser's FWHM determined from soybean oil measurements, a series of 2D Gaussian filter masks with unity peak values were created centered at the guide star positions with variances based on the measured FWHM. Once the chemical signature volumes were reconstructed with BS-IDT, these filters were applied to each IR beam reconstruction to select only the signatures. Each filtered volume was summed together and normalized by the sum of the Gaussian filters to reduce RI errors from filtering. The final stitched volume was shown in Fig. 5 to generate visualizations of the full *C. elegans* chemical signatures. For plotting the spectroscopic information, the individual IR beam illumination reconstructions were still used to prevent stitching errors from altering the quantitatively recovered fingerprint spectra.

## Biological sample preparation

Authenticated T24 and SW780 cells were obtained from the American Type Culture Collection (ATCC). The T24 cells were cultured in McCoy's 5 A media (Gibco, USA), and the SW780 cells were cultured in Gibco Roswell Park Memorial Institute (RPMI) media (Gibco, USA). All media were supplemented with 10% fetal bovine serum and 1% penicillin-streptomycin. Cell lines were incubated at 37 C° with 5% $CO_2$. Cells were treated with 50 μM oleic acid (Sigma, USA) for 6 h. Then the cells were washed by $H_2O$-based PBS once and then fixed in 10% neutral buffered formalin. After fixation, the cells were washed five times by $D_2O$-based PBS. Finally, the fixed cells were immersed in $D_2O$ PBS solutions and sandwiched by two pieces of Raman-grade 0.2 mm-thick $CaF_2$ glass.

*C. elegans* (daf-2 (e1370) mutant strain) were cultured on standard nematode growth medium (NGM) plates (5 mg/L cholesterol) with *Escherichia coli* (*E. coli*) OP50 at 20 °C using standard protocols[2]. Both daf-2 mutant *C. elegans* and *E. coli* OP50 were purchased directly from the Caenorhabditis Genetics Center at the University of Minnesota. After being cultured for 2–3 days, the worms were first detached from the NGM plate, washed by $H_2O$ PBS, and fixed by 10% formalin overnight at room temperature. Then, the worms were washed further 4 times to remove the *E. coli* OP50 with $D_2O$ PBS. Finally, the samples were sandwiched by two pieces of Raman-grade 0.2-mm-thick $CaF_2$ glass and immersed in $D_2O$ PBS.

## Software

Data were collected by customized MATLAB code. Data were processed by customized MATLAB code and Fiji ImageJ (Version:1.53 c).

## Reporting summary

Further information on research design is available in the Nature Portfolio Reporting Summary linked to this article.

## Data availability

All the data are available upon reasonable request to the corresponding authors (jxcheng@bu.edu(J.X.C.) and leitian@bu.edu(L.T.)). All bladder cancer cell reconstruction data and *C. elegans* reconstruction data related to displayed items are available at https://github.com/buchenglab/BS-IDT_Data. Source data are provided in this paper. Source data are provided with this paper.

## Code availability

The codes are available at https://github.com/buchenglab/BS-IDT.

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

## Acknowledgements

This work is supported by a grant from Daylight Solutions to J.Z., J.B.Z., L.T., and J.X.C.

## Author contributions

J.Z., A.M., L.T., and J.X.C. proposed the idea of BS-IDT. J.X.C. and L.T. supervised the research team and revised the manuscript. J.Z. proposed the idea of a pulsed ring laser illumination system, designed the BS-IDT system and the imaging experiments, built the BS-IDT imaging system, prepared the biological samples, performed hyperspectral imaging experiments, and wrote the first draft. A.M. developed the BS-IDT and the BS-DPC imaging model, led the coding realizations of the imaging model and data processing, assisted with the draft writing, and revised the manuscript. J.Z. and A.M. performed the data acquisition for chemical imaging and system characterizations. A.M., J.Z., and J.B.Z. all contributed to the data processing. J.Z., Z.Q.S, B.W., X.C.L., and H.B.Z. developed the ring laser illumination system. H.B.Z. led the manufacture of the ring laser illumination system. F.K.C. and Z.C.C. cultured the cells used in this manuscript. Y.W.Z. cultured the *C.elegans* used in this manuscript. Y.H.X. assisted with the data acquisition of the system characterizations and hyperspectral chemical imaging experiments. All authors contributed to the final creation of the manuscript.

## Competing interests

The authors declare no competing interests.
