## [Peer Review File · Nature Communications]

Bond-Selective Intensity Diffraction TomographyEditorial Note: Parts of this peer review file have been redacted as indicated to avoid any copy right infringement.

REVIEWER COMMENTS

Reviewer #1 (Remarks to the Author):

This study developed a modality that incorporates mid-infrared photothermal microscopy and diode laser-based intensity diffraction tomography for 3D hyperspectral imaging. This modality uses a lab-built laser ring array containing 16 diode lasers for probing. The authors showcase the applications of this new modality in imaging/quantifying lipids and protein in cancer cells, and volumetric chemical imaging on *C. elegans*. Compared with previous study using optical diffraction tomography, this method achieves a 40 times faster imaging speed, 3-fold increase in depth resolution, and 3 times larger field of view.

As a new non-interferometric computational MIP microscopy scheme for 3D bond-selective label-free imaging. The authors used the time-gated pump-probe MIP microscopy with the pulsed-laser-based intensity diffraction tomography, termed Bond-Selective Intensity Diffraction Tomography (BS-IDT).

The explanation at the beginning of results section is good enough for readers to understand the working principle. The limitation at this stage and perspectives are also well explained in the discussion. The results presented are noteworthy and the work is important to the field.

This technique has great potential as a fast 3D chemical imaging modality. This paper can be improved by addressing the following comments and suggestions.

1. In the introduction (line 96) and discussion (line 425), a specific number of spatial resolution was mentioned. It is hard to find the information on how to determine this number in this paper. It would be better to supply the information about the method.
2. In line 250, the temporal decay constant of 20 μs was mentioned. This number is based on C=O stretch vibration in the oil sample. If additional results about the C=O stretch vibration in biomolecules are provided, the number can support the next datasets.
3. In Figure 3, the volumes of lipid droplets were presented. It would be better to provide the method of how the LD volumes in the plot were measured.
4. As a vibrational imaging platform, utilizing Cell Silent Region from 1800cm^{-1} to 2800cm^{-1} for metabolic dynamic in living organism will enhance the impact of this new microscopy's applications. For example, imaging newly synthesized proteins and lipids signals can be very useful for researchers to study metabolic dynamics in cells and tissues (DOI: 10.1038/s41467-018-05401-3; doi.org/10.1038/s41592-020-0883-z). Or at least, the authors want to include the potential metabolic imaging applications in the discussion.
5. Figure 2b. It is unclear/misleading what the x-axis is. (Lines 241 & 250, e.g., 0.5 μs in the study), was the time spacing between each mid-IR pulses (e.g. 100 μs)? Are authors examining the RI decay (cooling) in the sample, as indicated in Line 254 that "temporal pulse spacing ... $\sim 100 \mu\text{s}$, allowing sufficient cooling". Maybe rephrase it for clarification.

6. It would be easier to see the decay of RI if RI changes are normalized and presented as percentages of the initial value. Also, extend the x-axis to 100 μ s, as the pulse spacing in the current study.

7. Line 281-282, the sentence "For each image ... 0.05 Hz acquisition rates" is confusing. Is the acquisition time \sim 20 μ s for each volume or each frame of images (e.g. each of Figure 3a1, a2 & a3)?

8. In the Introduction and Discussions, the authors stated that the lateral and axial resolutions to be 350 nm and 1.1 μ m, respectively. However, these are not clearly presented in the Results, except that the axial resolution of 1.1 μ m may be guessed from Figure 3 & 5. From where can the lateral resolution (350 nm) be told?

9. Line 327, "alternation" --- "alteration".

10. "0 μ m" to "0 μ m" (line 363).

11. Line 462, "significantly" --- "significant".

Reviewer #2 (Remarks to the Author):

Comments for authors

Report on the manuscript: Bond-Selective Intensity Diffraction Tomography

by Jian Zhao¹, Alex Matlock¹, Hongbo Zhu, Ziqi Song, Jiabei Zhu, Biao Wang, Fukai Chen, Yuewei Zhan, Zhicong Chen, Yihong Xu, Xingchen Lin, Lei Tian and Ji-Xin Cheng

The manuscript reports on a new optical method for fast and high-resolution volumetric imaging with chemical specificity using mid-IR photothermal microscopy previously developed by one of the authors' group (J. X. Cheng's group). The authors show several key advantages of the BS-IDT (Bond-Selective Intensity Diffraction Tomography) technique by imaging various biological cell samples. Fast and high-resolution 3D imaging with chemical bond specific information is indeed beneficial for understanding complex cellular structures. The experiments reported in the manuscript have been carefully performed and thoroughly discussed. The article is definitely a useful contribution to label-free cellular imaging. I therefore recommend publication of the article after the authors consider the following comments.

Major remarks:

1. Line 96 (also mentioned later in line 425): The authors mentioned high-resolution capabilities of BS-IDT with specified lateral resolution of ~ 350 nm and axial resolution of ~ 1.1 μ m. However, the authors did not present any data which showed the above specified resolution. It would be ideal to show an image obtained using BS-IDT and a line profile showing both the above-specified lateral and axial resolutions.

2. Line 147: The authors used a probe laser with a pulsed illumination instead of CW illumination to obtain a photothermal image. This is in contrast to standard visible photothermal imaging where mostly CW probe laser is used for imaging. It would be indeed beneficial for the broader community if the authors can comment on the difference in contrast (or SNR), resolution and any other constraints if CW probe illumination is used instead of pulsed illumination.

3. Figure 4: The figure demonstrates that higher image contrast in BS-IDT than BS-DPC according to the authors, however the line profiles do not show strong differences in these two imaging modalities. A quantitative analysis is required here. In addition, the contrast will depend on the excitation intensity and integration time. Therefore, image contrast (or SNR) needs to be quantified with considering both these parameters.

Minor remarks:

1. Line 38: The authors mentioned cellular function perturbation as one of the weaknesses of fluorescence microscopy. Although it is a general statement, a proper reference will justify this statement.

2. Line 221: The refractive index change due to the temperature change is in the order of $\sim 10^{-4}$ to 10^{-3} . J. X. Cheng's group previously demonstrated that instead of scattering-based photothermal imaging, temperature-induced fluorescence intensity fluctuation which is in the order of 10^{-2} , can enhance the photothermal sensitivity by two orders of magnitude. It would be nice to point out in the outlook if the current BS-IDT method can also be implemented in fluorescence-detected mid-IR photothermal microscopy (J. Am. Chem. Soc. 2021, 143, 30, 11490–11499).

3. Line 5-7: Is there any specific reason to use 16 diode lasers? Is there any limitations if less than 16 or higher than 16 lasers array is used?

4. References: in ref. 45, instead of "cm⁻¹", "cm⁻¹" and in ref. 53, instead of "LiveCaenorhabditis", "Live Caenorhabditis".

Reviewer #3 (Remarks to the Author):

Zhao et al. describe in their manuscript entitled "Bond-Selective Intensity Diffraction Tomography" a method for bond selective photothermal mid-infrared 3 dimensional imaging. The method is a combination of time-gated photothermal imaging in the mid-infrared region, which was developed earlier, and the method of intensity diffraction tomography (IDT), a computational imaging method that has been reported in the literature. Here, the IDT is implemented with 16 lasers that provide oblique illuminations for the imaging. The pulsed intensity of the 16 lasers is synchronised with the pulsed infrared pump laser and the camera to provide a complex sequence of images from which the 3D image can be reconstructed. The combination of these two schemes is demonstrated to provide 3-dimensional information about the distribution of chemical species of single cells, cell organelles and multicellular organisms.

This is a novel complex but powerful imaging technique, which allows for label-free 3 dimensional imaging with chemical resolution. The spatial and temporal resolution of this technique is superior to other available techniques. The authors clearly demonstrate mid-infrared photothermal character of the signals.

The manuscript is nicely written and the data is of very high quality and I congratulate the authors for this nice work. I have only smaller issues.

1) On page 2 line 53 the authors mention the cross-section for Raman scattering. I would assume that they refer to the Raman scattering cross-section and not the absorption cross-section.

2) The authors state in the methods section that they use 450 nm laser for the ring illumination with a CW power under 3W and an energy fluency of $0.2 \text{ pJ}/\mu\text{m}^2$ with pulses of 1 μs length. Given the arguments in the introduction concerning the Raman microscopy, it would be fair to compare the photon budget to Raman spectroscopy as well.

3) Refractive index changes of up to 3×10^{-3} are reported in the experiments. Given typical thermo-optic coefficients, I would estimate a temperature change of about 10 K or more. Could the authors comment on that?

4) I would think that the integration in eq. 1 is over d^3r .

Overall, this is a very nice piece of work and I can recommend it for publication in Nature Communications.

Point-by-point response

We greatly appreciate the referees for their positive reviews and constructive comments that we have received on our manuscript submitted to Nature Communications (Manuscript ID: NCOMMS-22-14397).

Please find our detailed response below. The original referee's comments are in blue-color italic font, while our responses are in black for accessible communication. The corresponding changes in our manuscript and supplementary information note are highlighted in yellow.

Figure 3 correction from the authors:

During the manuscript revision, we found that we confused the bladder cancer cell chemical imaging data between the off-resonance "1900 cm⁻¹" and the "IR-off" columns when making the **Figure 3** in the original manuscript. Therefore, in the original manuscript, those images in **Figure 3 d** should be labeled as "IR off" while the other images in **Figure 3 e** should be labeled as "1900 cm⁻¹". As illustrated in the above figure, we have corrected these issues in our revised manuscript. Please refer to the updated **Figure 3** in our revised manuscript. Here, both "1900 cm⁻¹" and "IR-off" chemical images were originally designed as control groups and expected to demonstrate similar low signal-to-noise ratios. The above issues do not affect any conclusions in our manuscript.

In addition to the abovementioned corrections, we further added a detailed legend for the box plot in **Figure 3 n** in the revised manuscript. We also revised the format of the manuscript according to the requirements of Nature Communications and updated the current address of the first author.

Reviewer #1 (Remarks to the Author):

*This study developed a modality that incorporates mid-infrared photothermal microscopy and diode laser-based intensity diffraction tomography for 3D hyperspectral imaging. This modality uses a lab-built laser ring array containing 16 diode lasers for probing. The authors showcase the applications of this new modality in imaging/quantifying lipids and protein in cancer cells, and volumetric chemical imaging on *C. elegans*. Compared with previous study using optical diffraction tomography, this method achieves a 40 times faster imaging speed, 3-fold increase in depth resolution, and 3 times larger field of view.*

As a new non-interferometric computational MIP microscopy scheme for 3D bond-selective label-free imaging. The authors used the time-gated pump-probe MIP microscopy with the pulsed-laser-based intensity diffraction tomography, termed Bond-Selective Intensity Diffraction Tomography (BS-IDT).

The explanation at the beginning of results section is good enough for readers to understand the working principle. The limitation at this stage and perspectives are also well explained in the discussion. The results presented are noteworthy and the work is important to the field.

This technique has great potential as a fast 3D chemical imaging modality. This paper can be improved by addressing the following comments and suggestions.

Author response:

We sincerely thank the reviewer for the comprehensive and positive evaluation of our manuscript.

1. *In the introduction (line 96) and discussion (line 425), a specific number of spatial resolution was mentioned. It is hard to find the information on how to determine this number in this paper. It would be better to supply the information about the method.*

8. *In the Introduction and Discussions, the authors stated that the lateral and axial resolutions to be 350 nm and 1.1 μm , respectively. However, these are not clearly presented in the Results, except that the axial resolution of 1.1 μm may be guessed from Figure 3 & 5. From where can the lateral resolution (350 nm) be told?*

Author response:

We thank the referee for the comments. We agree with the referee that related data should be presented to consolidate this statement. Both referee's comments **1** and **8** focus on the same topic regarding the resolution characterizations. Therefore, we respond to both comments with a single response. The lateral and axial resolution of ~ 350 nm and ~ 1.1 μm were evaluated by the experimental measurement of the 3D chemical imaging results. The above resolutions are consistent with the system's diffraction-limited resolution. Related experimental measurements are shown in **Figure R1** below.

Figure R1. Resolution characterizations using lipid chemical imaging result from bladder cancer cells. The lipid chemical imaging data used here are further improved by halo artifacts removal processing. **(a₁)** The image corresponds to a depth of $\sim -1.065 \mu\text{m}$. A selected lipid droplet is indicated by the white arrow shown in the inset. **(a₂)** Blue curve: extracted lateral profile along the yellow dashed line cross the selected lipid droplet in **(a₁)**; Red curve: Gaussian line shape fitting (FWHM: $\sim 349 \text{ nm}$, R square coefficient=0.99) for the main peak corresponding to the selected lipid droplet. **(b₁)** The image corresponds to the depth of $\sim -0.532 \mu\text{m}$. A selected lipid droplet is indicated by the white arrow shown in the inset. The orthogonal view of the selected lipid droplet is also demonstrated. "Z" indicates the depth direction. "X" and "Y" indicate lateral direction. **(b₂)** Blue curve: extracted depth profile from the selected lipid droplet's peak signal in **(b₁)**; Red curve: Gaussian line shape fitting (FWHM: $\sim 1.082 \mu\text{m}$, R square coefficient=0.95) for the main peak corresponding to the selected lipid droplet

We characterized the resolution of BS-IDT and demonstrated the results in **Figure R1**. We used the fixed bladder cancer T24 cells as the test bed. By performing depth-resolved chemical imaging at the 1745 cm^{-1} mid-IR wavenumber, we picked up two lipid droplets (**Figure R1 a₁, b₁**) to plot the lateral and axial line profiles (**Figure R1 a₂, b₂**). The Full-Width Half Maximum (FWHM) of the lateral line profile is $\sim 349 \text{ nm}$, and the FWHM of the axial line profile is $\sim 1.082 \mu\text{m}$. Here, we applied an additional halo artifact removal step to the chemical imaging data based on the work by Kandel *et al.* (**Ref A**).

(**Ref A**: "Real-time halo correction in phase contrast imaging," **Biomed. Opt. Express** 9, 623-635, 2018.)

We have added the above resolution characterization figures, results, and more details regarding the halo artifacts removal process as a new section, "**BS-IDT resolution characterization**," in the revised **Supplementary information**. Meanwhile, in the revised manuscript, we have added new statements in the second paragraph of the section "**3D chemical imaging, mid-IR fingerprint spectroscopy, and metabolic profiling of cancer cells**". The new statements are detailed below: "**We characterized the system resolution using the lipid features extracted from the absorption map at 1745 cm^{-1} (Supplementary information: "BS-IDT resolution characterization"). The measured lateral and axial FWHM linewidths are $\sim 349 \text{ nm}$ and $\sim 1.081 \mu\text{m}$, respectively.**".

2. In line 250, the temporal decay constant of 20 μs was mentioned. This number is based on C=O stretch vibration in the oil sample. If additional results about the C=O stretch vibration in biomolecules are provided, the number can support the next datasets.

Author response:

We thank the referee's comments. We would like to clarify the purpose of extracting the temporal decay constant and provide more experimental and simulation evidence.

Figure R2. Experimental measurement and numerical simulations of thermal decay. (a₁) BS-IDT measurement result for a PMMA bead immersed in D₂O. **(a₂)** Simulation results for a PMMA bead with the same size as **(a₁)** immersed in D₂O. **(b₁)** Simulation results for human fat beads of different volumes immersed in D₂O. **(b₂)** Simulation results for human fat beads of different volumes immersed in soybean oil. The exponential temporal decay time constants are obtained by exponential decay curve fitting. All the R square coefficients for the fittings are equal to or larger than 0.97.

1) For pump-probe MIP microscopy, we need to confirm that the exponential temporal decay time of samples under test is smaller than the period of the pump pulse (~100 μs). This will avoid continuously heating the sample by the pump pulses. Based on our team's previous publication, "**Background-Suppressed High-Throughput Mid-Infrared Photothermal Microscopy via Pupil Engineering, ACS Photonics, 2021**", we demonstrated that the system's temporal decay constant could be determined by scanning the temporal shift between pump and probe pulses for a pre-defined sample. This decay time increases with the microscopic objects' sizes. Small objects' heat dissipates faster than that of large objects. Furthermore, samples immersed in water demonstrate a shorter decay time compared to those immersed in oil.

In our BS-IDT manuscript, we obtained the temporal decay time of oil film ($\sim 20 \mu\text{s}$) with a similar method. Our test bed is based on a large oil film, which is supposed to be exposed to the largest infrared beam absorption volume and demonstrate the longest temporal decay time among all samples under test. The biological samples' microscopic features in our manuscript are much smaller than the soybean oil film. Also, these samples are immersed in phosphate-buffered-saline based heavy water instead of oil. Therefore, the temporal decay constants of biological samples in our manuscript are smaller than the measured decay time using the oil film ($\sim 20 \mu\text{s}$). Considering the period of our pump pulse ($\sim 100 \mu\text{s}$), our samples under test all have enough cooling time.

2) We performed experimental measurements and numerical simulations, as shown in **Figure R2** above, to provide additional evidence. We measured the thermal decay of a polymethyl methacrylate (PMMA) bead immersed in heavy water and performed the corresponding simulation as a comparison (**Figures R2 a₁** and **a₂**). Here, PMMA bead's chemical composition contains the C=O bond. A consistent thermal decay process can be observed for both experimental and numerical tests (**Figures R2 a₁** and **a₂**). The decay time constant is much faster than oil film, with a time shorter than $1 \mu\text{s}$. We further performed numerical simulations for human fat (biomolecule with the C=O bond) beads with varying volumes and immersed in different media. The bead is located in the center of a sphere composed of either heavy water or soybean oil. The diameter of the sphere is 40 times larger than the bead's diameter. The initial temperature of the sphere and the bead were set as 293 K and 298 K, respectively. As shown in **Figure R2**, the chemical compositions of the beads do not play a significant role in the decay time, while the larger size can indeed increase the temporal decay constant. In addition, beads immersed in heavy water demonstrate almost two times smaller temporal decay constant than those immersed in oil. For any cases, the decay time is less than or around $8 \mu\text{s}$. The above results further validate that our temporal decay constant extracted from the oil film sample can safely determine the decay time for the BS-IDT system since all our biological samples are immersed in a heavy water environment.

3) We have added the data shown in **Figure R2** and related statements as a new "**Heat dissipation measurement and simulation**" section in the revised **Supplementary information**.

We also added new statements in first paragraph in the section of "**BS-IDT System Characterization**" in the revised manuscript. The new statements are as below: "**We performed experimental measurement and numerical simulations for particles with sizes comparable to organelles in cells (Supplementary information: "Heat dissipation measurement and simulation").** These particles demonstrate much faster temporal decay processes compared to the oil film."

3. In Figure 3, the volumes of lipid droplets were presented. It would be better to provide the method of how the LD volumes in the plot were measured.

Author response:

We thank the referee's comment. In our previous submission, we provided the lipid droplet volume quantification method in the "**Lipid quantification**" section in the original **Supplementary information** notes. We agree that it would be more clear to add related descriptions in the main text of the manuscript. For this purpose, we add the following statements in the revised manuscript: "**We calculated the lipid droplet volume by converting the total voxels into true spatial dimensions (Supplementary information: "Lipid quantification").**" Please find the above revision in the last paragraph of the section of "**3D**

chemical imaging, mid-IR fingerprint spectroscopy, and metabolic profiling of cancer cells" in the revised manuscript.

4. As a vibrational imaging platform, utilizing Cell Silent Region from 1800cm^{-1} to 2800cm^{-1} for metabolic dynamic in living organism will enhance the impact of this new microscopy's applications. For example, imaging newly synthesized proteins and lipids signals can be very useful for researchers to study metabolic dynamics in cells and tissues (DOI: 10.1038/s41467-018-05401-3; doi.org/10.1038/s41592-020-0883-z). Or at least, the authors want to include the potential metabolic imaging applications in the discussion.

Author response:

We thank the referee for the suggestions. We agree that metabolic dynamic imaging would be an important application of our BS-IDT method. To this end, we have added the following statements and a new reference **Ref 64** ("Optical imaging of metabolic dynamics in animals", **Nature Communications**, 2018') in the revised manuscript:

"Fourth, BS-IDT can be applied to IR metabolic dynamic imaging of living organisms, covering single-cell-level and tissue-level metabolic activity profiling. Such metabolic imaging can be implemented with the assistance of the IR-active vibrational probes and by extending the spectral coverage to the cell-silent window ($1800\text{ cm}^{-1} - 2700\text{ cm}^{-1}$). For example, IR tags, such as the azide group, ^{13}C -edited carbonyl bond, and deuterium-labeled probes²⁵, can be exploited to investigate various metabolic activities simply by adding another QCL laser module to cover both the fingerprint region and the cell-silent region. BS-IDT imaging of IR tags with a sub-micrometer resolution, high speed, and rich spectral information would contribute to unraveling the mechanisms of many biological processes^{25,64}".

Please find the above revision in the fourth paragraph of the revised manuscript's "Discussion" section.

5. Figure 2b. It is unclear/misleading what the x-axis is. (Lines 241 & 250, e.g., $0.5\ \mu\text{s}$ in the study), was the time spacing between each mid-IR pulses (e.g. $100\ \mu\text{s}$)? Are authors examining the RI decay (cooling) in the sample, as indicated in Line 254 that "temporal pulse spacing ... $\sim 100\ \mu\text{s}$, allowing sufficient cooling". Maybe rephrase it for clarification.

Author response:

We thank the referee's comments. The delay time in the x-axis of **Figure 2b** corresponds to the temporal shift between the center of the probe pulse and the center of the pump pulse. Therefore, it is not the time spacing between each mid-IR pulses. It is the time spacing between the visible probe pulse and the mid-IR pump pulse. Since there exists a heat dissipation process, the response induced by the photothermal effect starts to degrade with a large delay time. By fitting the curve with this degradation process, we can estimate the decay time using a sample whose heat dissipation speed is much slower than the biological sample under investigation. Please refer to our response to **comment 2** for more details.

To clarify the definition of the pump-probe delay time, we make the following revisions in the manuscript.

- 1) We revised **Figure 2 b**): the x-axis of **Figure 2 b**) is revised as "Pump-Probe Pulse Delay Time (μs)".
- 2) In the first paragraph of the section "BS-IDT System Characterization", we added the following statements:"..... between the pump and probe laser pulses, and the mid-IR wavenumber. Here, the delay time is defined as the temporal shift between the probe pulse's center and the pump pulse's center."

6. *It would be easier to see the decay of RI if RI changes are normalized and presented as percentages of the initial value. Also, extend the x-axis to 100 μ s, as the pulse spacing in the current study.*

Author response:

We thank the referee for the suggestion. We have revised **Figure 2 b)** by adding a new axis labeled "Relative Variation (%)". We normalized the RI changes by the maximum RI value to obtain the relative variations. Please find the updated **Figure 2 b)** in the revised manuscript. Our experimental data do not extend to $\sim 100 \mu$ s. Being limited by the highly customized ring-laser electronic system, we have technical difficulties in keeping a stable timing synchronization among pump pulse, probe pulse, and camera frame rate if we detune the time delay between the pump pulse and probe far beyond 50μ s. For the current data, we already have enough data to extract the decay variation curve. Please refer to our response to **comment 2** for more details regarding the thermal decay characterizations.

7. *Line 281-282, the sentence "For each image ... 0.05 Hz acquisition rates" is confusing. Is the acquisition time $\sim 20 \mu$ s for each volume or each frame of images (e.g. each of Figure 3a1, a2 & a3)?*

Author response:

We thank the referee's comments. The 0.05Hz acquisition rates correspond to 20 seconds. This acquisition time (~ 20 seconds/0.05Hz) is for a complete 3D volumetric chemical imaging dataset. A complete 3D imaging dataset contains all the 2D images, each of which corresponds to different depths. In our manuscript, we just extract 3 sample images from a single 3D imaging dataset for demonstration, for example, **Figure 3 a₁, a₂ and a₃**.

9. *Line 327, "alternation" --- "alteration".*

Author response:

We thank the referee for pointing out this issue. We have corrected it as "alteration".

10. *"0 μ m" to "0 μ m" (line 363).*

Author response:

We thank the referee for pointing out this issue. We have corrected the caption.

11. *Line 462, "significantly" --- "significant".*

Author response:

We thank the referee for pointing out this issue. We have corrected it as "significant".

Reviewer #2 (Remarks to the Author):

The manuscript reports on a new optical method for fast and high-resolution volumetric imaging with chemical specificity using mid-IR photothermal microscopy previously developed by one of the authors' group (J. X. Cheng's group). The authors show several key advantages of the BS-IDT (Bond-Selective Intensity Diffraction Tomography) technique by imaging various biological cell samples. Fast and high-resolution 3D imaging with chemical bond specific information is indeed beneficial for understanding complex cellular structures. The experiments reported in the manuscript have been carefully performed and thoroughly discussed. The article is definitely a useful contribution to label-free cellular imaging. I therefore recommend publication of the article after the authors consider the following comments.

Author response:

We would like to thank the referee for the thoughtful comments, valuable suggestions, and positive evaluations.

Major Remark

1. Line 96 (also mentioned later in line 425): The authors mentioned high-resolution capabilities of BS-IDT with specified lateral resolution of ~ 350 nm and axial resolution of ~ 1.1 μm . However, the authors did not present any data which showed the above specified resolution. It would be ideal to show an image obtained using BS-IDT and a line profile showing both the above-specified lateral and axial resolutions.

Author response:

We would like to thank the referee for the comments. We agree with the referee that related data should be presented to consolidate this statement. The lateral and axial resolution of ~ 350 nm and ~ 1.1 μm were evaluated by the experimental measurement of the 3D chemical imaging results. The above resolutions are consistent with the system's diffraction-limited resolution. Related experimental measurements are shown in **Figure R3** below.

We characterized the resolution of BS-IDT and demonstrated the results in **Figure R3**. We used the fixed bladder cancer T24 cells as the test bed. By performing depth-resolved chemical imaging at the 1745 cm^{-1} mid-IR wavenumber, we picked up two lipid droplets (**Figure R3 a₁, b₁**) to plot the lateral and axial line profiles (**Figure R3 a₂, b₂**). The Full-Width Half Maximum (FWHM) of the lateral line profile is ~ 349 nm, and the FWHM of the axial line profile is ~ 1.082 μm . Here, we applied an additional halo artifact removal step to the chemical imaging data based on the work by Kandel *et al.* (**Ref A**).

(**Ref A**: "Real-time halo correction in phase contrast imaging," **Biomed. Opt. Express** 9, 623-635, **2018**.)

We have added the above resolution characterization figures, results, and more details regarding the halo artifacts removal process as a new section, "**BS-IDT resolution characterization**," in the revised **Supplementary information**. Meanwhile, in the revised manuscript, we have added new statements in the second paragraph of the section "**3D chemical imaging, mid-IR fingerprint spectroscopy, and metabolic profiling of cancer cells**". The new statements are detailed below: "We characterized the system resolution using the lipid features extracted from the absorption map at 1745 cm^{-1} (**Supplementary information**). The measured lateral and axial FWHM linewidths are ~ 349 nm and ~ 1.081 μm , respectively."

Figure R3. Resolution characterizations using lipid chemical imaging result from bladder cancer cells. The lipid chemical imaging data used here are further improved by halo artifacts removal processing. **(a₁)** The image corresponds to a depth of $\sim -1.065 \mu\text{m}$. A selected lipid droplet is indicated by the white arrow shown in the inset. **(a₂)** Blue curve: extracted lateral profile along the yellow dashed line cross the selected lipid droplet in **(a₁)**; Red curve: Gaussian line shape fitting (FWHM: $\sim 349 \text{ nm}$, R square coefficient=0.99) for the main peak corresponding to the selected lipid droplet. **(b₁)** The image corresponds to the depth of $\sim -0.532 \mu\text{m}$. A selected lipid droplet is indicated by the white arrow shown in the inset. The orthogonal view of the selected lipid droplet is also demonstrated. "Z" indicates the depth direction. "X" and "Y" indicate lateral direction. **(b₂)** Blue curve: extracted depth profile from the selected lipid droplet's peak signal in **(b₁)**; Red curve: Gaussian line shape fitting (FWHM: $\sim 1.082 \mu\text{m}$, R square coefficient=0.95) for the main peak corresponding to the selected lipid droplet

2. Line 147: The authors used a probe laser with a pulsed illumination instead of CW illumination to obtain a photothermal image. This is in contrast to standard visible photothermal imaging where mostly CW probe laser is used for imaging. It would be indeed beneficial for the broader community if the authors can comment on the difference in contrast (or SNR), resolution and any other constraints if CW probe illumination is used instead of pulsed illumination.

Author response:

We would like to thank the reviewer for this comment. We agree that it is necessary to discuss this topic, and we have made the corresponding revisions to our manuscript.

First, using a pulsed probe beam in IR photothermal widefield microscopy has been demonstrated by different groups in recent years, such as "*Ultrafast chemical imaging by widefield photothermal sensing of infrared absorption*, **Science Advances**, 5, eaav7127, **2019**", "*Quantitative phase imaging with molecular vibrational sensitivity*, **Optics Letters**, 44, 3729, **2019**", and "*Bond-selective transient phase imaging via sensing of the infrared photothermal effect*, **Light: Science & Applications**, 8, 116, **2019**". For our BS-IDT widefield system, we use a pump-probe detection scheme similar to those deployed in the abovementioned works.

The reason for using point-scanning and CW probe laser in standard photothermal imaging can be understood below. The point-scanning-based method detects the modulation of the visible probe light using the high-speed photodetector and extracts the weak signal (<1% modulation depth) using the lock-in amplifier. The speed and sensitivity are high enough to track fast-varying weak signals directly. Therefore, this type of system can use CW visible probe beams.

In contrast, pulsed probe beam detections are preferable for widefield IR photothermal microscopy. The main reasons are 1) the camera's slow speed; 2) Camera detection lacks a lock-in amplifier. Specifically speaking, most regular cameras' frame rates can hardly reach tens of kilohertz, thereby being difficult to capture the real-time modulation of the probe light. Instead, the cameras record the integrated probe light over a long time window determined by the low frame rate. Meanwhile, camera detection struggles to extract those weak signals without the lock-in amplifier. Taking our BS-IDT system as an example, the pump pulse is operated at 10kHz with a $\sim 1 \mu\text{s}$ pulse duration while the camera's frame rate is merely 100Hz. Within one frame (~ 0.01 seconds), the probe light is modulated with a low modulation depth (<1%) by the pump pulses. There are only ~ 100 pump pulses with a short duration of $\sim 1 \mu\text{s}$ participating in this modulation. If using a CW probe laser, a small fraction of the probe light is modulated while the camera integrates all the CW probe light within one frame. Therefore, for CW probe light, the contrast and signal-to-noise ratio will be quite low, resulting in low sensitivity. Despite that disadvantage, the resolution should not be affected using a CW probe light.

Yet, CW probe beam could still be applied to realize high contrast and signal-to-noise ratio imaging even for widefield IR photothermal microscopy if an ultrafast camera with a frame rate of tens of kilohertz and a high sensitivity is applied. It means that the camera is fast enough to capture the instant modulation from the pump pulse (e.g., pump pulse for BS-IDT: $\sim 10\text{kHz}$). The ultrafast camera could record the real-time photothermal modulated CW probe light in this case. High-contrast and high signal-to-noise ratio images should be feasible for this scenario.

Please find the following revisions in the revised manuscript.

1) New statements in the last paragraph of the section "BS-IDT principle, instrumentation, and image reconstruction": "For widefield MIP microscopy, the application of a pulsed probe laser here increases the contrast and sign-to-noise ratio, considering the slow frame rate of our camera^{30,31}."

2) New statements in the third paragraph of the section "Discussion": "In addition, the BS-IDT could potentially deploy the visible probe illumination under CW mode to simplify the timing scheme further. To this end, an ultrafast camera with a frame rate matching the repetition rate of the pump laser pulse is required."

3. Figure 4: The figure demonstrates that higher image contrast in BS-IDT than BS-DPC according to the authors, however the line profiles do not show strong differences in these two imaging modalities. A quantitative analysis is required here. In addition, the contrast will depend on the excitation intensity and integration time. Therefore, image contrast (or SNR) needs to be quantified with considering both these parameters.

Author response:

We would like to thank the reviewer for this comment. We agree that further elaborations on this BS-IDT and BS-DPC comparison are necessary.

Figure R4. Comparison of BS-IDT and BS-DPC microscopy. (a₁-a₃) BS-IDT cold and chemical 3D RI reconstruction results sectioned at 0 μm depth. Sample: fixed mitotic bladder cancer cells in D₂O PBS. (b₁-b₃) BS-DPC cold and chemical 2D phase reconstruction results of the same sample. For images in a) and b), each row corresponds to the same mid-IR radiation. c₁) and c₂) are line profiles extracted from the chemical imaging results of BS-IDT and BS-DPC. c₁) Yellow-color profile line is extracted from the yellow dash line area of a₂). Blue-color profile line is extracted from the blue dash line area of b₂). c₂) Red-color profile line is extracted from the red dash line area of a₃). Green-color profile line is extracted from the green dash line area of b₃). d₁) and d₂) are line profiles extracted from the "cold" imaging results of BS-IDT and BS-DPC. d₁) Yellow-color profile line is extracted from the yellow dash line area of a₁). Blue-color profile line is extracted from the blue dash line area of b₁). d₂) Red-color profile line is extracted from the red dash line area of a₁). Green-color profile line is extracted from the green dash line area of b₁).

First of all, in our original manuscript, we realized there exist the following issues to be improved, although these issues do not affect our main conclusions. **1)** We tried to extract the refractive index (RI) map from the BS-DPC's phase map in order to make a direct comparison. However, this 2D phase-to-RI conversion process is non-trivial and might not estimate 2D RI maps accurately from BS-DPC results. **2)** We did not select the positions of the lines properly, which misses the high-contrast areas. We explain each point in detail as below.

The 2D-limited BS-DPC method inherently recovers the integrated phase map through the object's volume at each pixel. This is different from BS-IDT. BS-IDT can recover the 3D RI map through the object's volume. In our original manuscript, we tried to recover the RI map from the BS-DPC's phase map results by assuming the object matched the objective's depth of field. The purpose is to directly compare BS-DPC and BS-IDT under the same scale. However, it is still challenging to accurately recover RI maps from phase-contributing features outside the depth of field. For now, it is hard to solve this issue. Therefore, we decide to abandon the efforts of converting BS-DPC's phase map to RI map. Instead, we compare the BS-DPC's phase map with the BS-IDT's RI map in the revised manuscript. On the basis of the BS-DPC's phase map and BS-IDT's RI map, we carefully re-select the line profiles that cross the high-contrast areas of the cell's main feature. We replot the line profiles for both chemical imaging results and cold imaging results. To this end, we revised **Figure 4** in the manuscript. We also demonstrate the revised **Figure 4** here. Please see **Figure R4** above.

Based on **Figure R4**, it is clear that BS-IDT demonstrates a much higher contrast than BS-DPC. The line profiles are consistent with the visual inspections of the imaging results. BS-IDT's high contrast originates from the algorithm itself based on comparing the cold imaging reconstructions. Here, BS-IDT and BS-DPC

are using exactly the same raw dataset that maintains the same parameters, such as integration time and pump-probe laser intensities. Yet, the signal-to-noise ratios demonstrated by the imaging reconstructions are the algorithm-recovered signal-to-noise ratio that is limited by the performance of the imaging reconstruction algorithm.

Based on the abovementioned points, we have made the following revisions to the section "**Comparison between 3D BS-IDT and 2D bond-selective differential phase contrast imaging**" in the revised manuscript.

1) Replace the previous BS-DPC RI map with BS-DPC phase map in **Figure 4**. Re-extract line profiles from all the chemical and cold imaging results of BS-IDT and BS-DPC based on BS-IDT RI map and BS-DPC phase map. Please find these updates in the revised manuscript's **Figure 4**.

2) We revised the corresponding statements. We emphasized the difference between BS-IDT and BS-DPC originates from the algorithms. We discussed the difficulties of converting BS-DPC phase into BS-DPC RI map as well as the advantages of estimating temperature variations using RI maps. Please find the highlighted sentences in the revised manuscript.

3) In the supplementary note, we delete the descriptions of converting BS-DPC phase map into BS-DPC RI map.

Minor Remarks

1. Line 38: The authors mentioned cellular function perturbation as one of the weaknesses of fluorescence microscopy. Although it is a general statement, a proper reference will justify this statement.

Author response:

We thank the referee's suggestions. We have added the following reference (Ref 2 in the revised manuscript) for this purpose: "*Fluorescent Probes for Lipid Rafts: From Model Membranes to Living Cells*, **Chemistry & Biology**, 21, 97, 2014".

2. Line 221: The refractive index change due to the temperature change is in the order of $\sim 10^{-4}$ to 10^{-3} . J. X. Cheng's group previously demonstrated that instead of scattering-based photothermal imaging, temperature-induced fluorescence intensity fluctuation which is in the order of 10^{-2} , can enhance the photothermal sensitivity by two orders of magnitude. It would be nice to point out in the outlook if the current BS-IDT method can also be implemented in fluorescence-detected mid-IR photothermal microscopy (J. Am. Chem. Soc. 2021, 143, 30, 11490–11499).

Author response:

We thank the referee for the suggestions.

Cheng's team and Simpson's team (J. Am. Chem. Soc. 2021, 143, 29, 10809–10815) simultaneously demonstrated similar works by exploiting the thermosensitive fluorescence dyes to enhance photothermal sensitivities. However, this concept cannot be directly applied to BS-IDT framework. The main reason is that BS-IDT assumes coherent illumination while fluorescence emission is inherently incoherent. Incoherent fluorescence imaging violates the assumption of our BS-IDT models and eliminates interference.

3. Line 5-7: Is there any specific reason to use 16 diode lasers? Is there any limitations if less than 16 or higher than 16 lasers array is used?

Author response:

We thank the referee's comment. The number of diode lasers is not restricted to 16. More lasers can benefit the imaging quality but reduce the imaging speed as well as increase the system cost. Fewer lasers

can also work but bring in degraded imaging quality. Compromising imaging quality, speed, and system cost, we decided to use 16 lasers.

4. References: in ref. 45, instead of "cm-1", "cm-1" and in ref. 53, instead of "LiveCaenorhabditis", "Live Caenorhabditis".

Author response:

We thank the referee for the comment. Related issues are corrected. Please check Ref 46 and Ref 54 in the revised manuscript.

Reviewer #3 (Remarks to the Author):

Zhao et al. describe in their manuscript entitled "Bond-Selective Intensity Diffraction Tomography" a method for bond selective photothermal mid-infrared 3 dimensional imaging. The method is a combination of time-gated photothermal imaging in the mid-infrared region, which was developed earlier, and the method of intensity diffraction tomography (IDT), a computational imaging method that has been reported in the literature. Here, the IDT is implemented with 16 lasers that provide oblique illuminations for the imaging. The pulsed intensity of the 16 lasers is synchronised with the pulsed infrared pump laser and the camera to provide a complex sequence of images from which the 3D image can be reconstructed. The combination of these two schemes is demonstrated to provide 3-dimensional information about the distribution of chemical species of single cells, cell organelles and multicellular organisms.

This is a novel complex but powerful imaging technique, which allows for label-free 3 dimensional imaging with chemical resolution. The spatial and temporal resolution of this technique is superior to other available techniques. The authors clearly demonstrate mid-infrared photothermal character of the signals.

The manuscript is nicely written and the data is of very high quality and I congratulate the authors for this nice work. I have only smaller issues.

Author response:

We sincerely thank the referee for the positive evaluation, detailed comments, and constructive feedback.

1) On page 2 line 53 the authors mention the cross-section for Raman scattering. I would assume that they refer to the Raman scattering cross-section and not the absorption cross-section.

Author response:

We would like to thank the referee for identifying this issue. We have corrected it as "scattering cross-section". Related revision is highlighted in the second paragraph of "Introduction" section.

2) The authors state in the methods section that they use 450 nm laser for the ring illumination with a CW power under 3W and an energy fluency of 0.2 pJ/μm² with pulses of 1 μs length. Given the arguments in the introduction concerning the Raman microscopy, it would be fair to compare the photon budget to Raman spectroscopy as well.

Author response:

We thank the referee's suggestions. We agree that it would be necessary to make a detailed comparison, and we have revised our manuscript for this purpose.

First of all, we would like to argue that there exists a fundamental difference between the BS-IDT and Raman or coherent Raman imaging regarding the illumination method (**Figure R5**). The beam intensity on the sample for Raman or coherent Raman microscope is about 10⁸ higher than the probe beam intensity on the sample for our BS-IDT based on recently published literature, which explains underpin BS-IDT's advantages over Raman microscopes for low photodamage.

[redacted])

Next, we would like to make a detailed explanation for the origin of BS-IDT's low probe beam intensity in the sample.

- 1)** BS-IDT is based on widefield imaging that does not require a tight focusing of the probe beam.
- 2)** BS-IDT's probe beam is delivered by a multimode optical fiber and propagates through a glass diffuser. This type of configuration makes the illumination beam diverge very fast, which further reduces the light intensity on the sample. As shown in **Figure R5 a₂**, the power that is effectively incident on the sample ($\sim 100 \mu\text{m}$ in diameter) is only a small fraction of the total power since the illumination area has a diameter of $\sim 4 \text{ cm}$.
- 3)** Although BS-IDT's probe beam has a high power of 3 W under continuous wave (CW) mode, our chemical imaging must be performed under pulse mode with a low duty cycle (pulse width: $\sim 1 \mu\text{s}$, repetition rate: 10kHz, duty cycle: ~ 0.01). The probe beam power under pulsed mode is about $\sim 30 \text{ mW}$. Especially, most of this power is not incident on the sample.
- 4)** The estimated intensity of probe beam incident on the sample is $\sim 2 \times 10^{-6} \text{ mW}/\mu\text{m}^2$ for BS-IDT. In contrast, the light intensity of beam incident on the sample for confocal Raman or coherent Raman microscope is orders of magnitude higher than BS-IDT.

For example, we estimate the light intensity incident on the sample for the Raman microscope (**Figure R5 b₂**) used in **Ref A** (bacterial classification), **Ref B** (volumetric Raman imaging), and **Ref C** (investigations of the Raman microscope). These works all use the same commercially available confocal Raman microscope (Alpha 300R+, WITec, Germany). The confocal Raman microscopy has to focus the incident beam into a tiny spot ($< 0.65 \mu\text{m}$) in the sample and perform a point scanning imaging (**Figure R5 b₁**), which generates a super high light intensity. Based on the data in **Ref B** (wavelength: $\sim 532 \text{ nm}$, power in sample: $\sim 40 \text{ mW}$), the light intensity is about $1 \times 10^2 \text{ mW}/\mu\text{m}^2$ for the confocal Raman microscope used in **Ref B**.

Similarly, we also estimate the light intensity of the stimulated Raman scattering (SRS) microscope demonstrated in **Ref D (Figure R5 b₃)**. This SRS microscope also relies on point-scanning design and delivers ~15 mW of pump beam power and ~75 mW of Stokes beam power into the sample. The pump beam power is measured to generate the SRS chemical image. If we merely consider the light intensity of the pump beam, it is about $\sim 5 \times 10^2 \text{ mW}/\mu\text{m}^2$.

Therefore, comparing the light intensity incident on the sample between BS-IDT and Raman or coherent Raman microscope, BS-IDT significantly reduces the photodamage risk. This is understandable. On the one hand, Raman scattering is a quite weak process that requires powerful light intensity; coherent Raman scattering involves a third-order nonlinear optical process that needs very high light intensity. On the other hand, BS-IDT relies on a linear absorption process with orders of magnitude larger cross-section than Raman scattering and probes the sample using a mature but efficient visible widefield imaging method.

We have made the following revisions to the revised manuscript.

1) In the "**Discussion**" section, we added the following statements and new references 58 and 59 to the second paragraph: *Benefiting from the widefield imaging scheme with a fast-diverging illumination design, the probe beam intensity incident on the sample is $\sim 2 \times 10^{-6} \text{ mW}/\mu\text{m}^2$, eight orders of magnitude lower than Raman⁵⁸ or coherent Raman microscope⁵⁹.*

2) In the section of "**Methods/Instrumentation**", we add the following statement: *"For all the chemical imaging experiments, the probe laser output power under pulsed mode is ~30 mW based on the ~0.01 duty cycle", "The probe beam illumination area on the sample has a diameter of around 4 cm".*

3) Refractive index changes of up to 3×10^{-3} are reported in the experiments. Given typical thermo-optic coefficients, I would estimate a temperature change of about 10 K or more. Could the authors comment on that?

Author response:

We thank the referee for this comment. The temperature change depends on the thermal expansion coefficient of the specific materials. Taking the water and the vegetable oil as examples, the thermal expansion coefficients of the water and oil at room temperature are $\sim 2 \times 10^{-4} \text{ K}^{-1}$, and $\sim 7 \times 10^{-4} \text{ K}^{-1}$, respectively. Therefore, for a refractive index change of $\sim 3 \times 10^{-3}$, the temperature changes for water and oil are $\sim 15 \text{ K}$ and $\sim 4 \text{ K}$, respectively. The temperature change can either be higher or lower than 10 K depending on the sample properties. Hence, the referee's estimation of the temperature change is generally correct.

4) I would think that the integration in eq. 1 is over d^3r' .

Author response:

We thank the reviewer for identifying this issue. We have corrected it in equation (1) as d^3r' . Please find the highlighted equation (1) in the revised manuscript.

Overall, this is a very nice piece of work and I can recommend it for publication in Nature Communications.

REVIEWERS' COMMENTS

Reviewer #1 (Remarks to the Author):

The authors have addressed all my concerns and I have no further questions. I support it for publication.

Reviewer #2 (Remarks to the Author):

Comments for authors

Report on the manuscript: Bond-Selective Intensity Diffraction Tomography

by Jian Zhao¹, Alex Matlock¹, Hongbo Zhu, Ziqi Song, Jiabei Zhu, Biao Wang, Fukai Chen, Yuewei Zhan, Zhicong Chen, Yihong Xu, Xingchen Lin, Lei Tian and Ji-Xin Cheng

The revised manuscript has thoroughly discussed the lateral and axial resolution of BS-IDT method, which was one of my major important remarks. The authors have carefully considered each of my inquiries and answered them in details. I appreciate the author's such an effort for in-depth discussion. I therefore recommend publication of the article as in its current form.

Reviewer #3 (Remarks to the Author):

I thank the authors very much for revising the manuscript and answering to my comments. Again, this is a valuable contribution to the field label-free cellular imaging and with the modifications and corrections carried out, the manuscript can now be recommended for publication.

Response to referees' comments

We greatly appreciate all the referees for their positive reviews and great efforts in helping us improve our manuscript that Nature Communications has accepted. We believe the publication of our bond-selective intensity diffraction tomography work will make new contributions to the development of label-free chemical imaging and benefit members working in related areas.